# Ternary Momentum For Quantized Training

**Noga Bar**[*] **& Amit Attia**
*Tel Aviv University*

**Michal Moshkovitz & Dotan Di-Castro**
*Bosch Research*

**Reviewed on OpenReview:** *https://openreview.net/forum?id=A3mVmPlahU*

## Abstract

Quantization enables efficient inference on resource-limited devices, yet training still depends on high-precision gradients and optimizer states. We address this gap by introducing stochastic ternary momentum, a fully quantized optimizer that operates with quantized parameters, ternary gradient information, and enables ternary momentum states for stable and memory efficient quantized optimization. Our method replaces deterministic and full-precision updates with integer-valued updates driven by stochastic sampling, ensuring that expected updates match standard momentum while maintaining strict memory constraints. It eliminates re-quantization overhead and preserves quantization consistency throughout training. We establish theoretical convergence guarantees of our ternary momentum method for convex objectives over bounded integer domains and for non-convex objectives over unbounded integer domains. Experiments on vision and language tasks demonstrate that our approach retains strong performance while reducing optimizer memory by 95% compared to full-precision, advancing the feasibility of fully quantized training.

## 1 Introduction

Quantization has emerged as a critical technique for deploying deep learning models on resource-constrained devices, as it reduces both memory footprint and computational cost during inference (Courbariaux et al., 2016; Han et al., 2016; Wang et al., 2019b; Jacob et al., 2018; Frantar et al., 2023; Li et al., 2016). However, quantization is typically applied only at deployment, while training still requires substantial memory and high-precision computations, particularly for large models where gradients and optimizer states must be stored (Rajbhandari et al., 2020). The gap between training and deployment requirements becomes more significant when using extremely low-bit-width quantized models, where the difference in resource demands is most pronounced. Successfully addressing the challenge of efficient training holds immense promise: it would unlock the ability to train larger models on devices with limited memory.

When training quantized models, there are four primary sources of memory usage: the *parameters and activations*, which are used during both training and inference, and the *gradients* and *optimizer states*, which are required only during training. In this work, our focus is on memory efficiency during training, specifically low bit-width optimizer states with only ternary gradient information to allow low-resources quantized optimization.

The use of *quantized gradients* is a natural line of research toward memory-efficient training. In this scenario, rather than relying on full precision availability of the gradients, only partial information about them is available. We focus on the case of ternary or binary values that provide extremely coarse approximations of the true gradient. Such estimators have been studied mainly in the context of communication efficiency for distributed learning rather than memory reduction (Wen et al., 2017; Sun et al., 2023; Bernstein et al., 2018;

---

*Noga conducted this work during an internship at Bosch.

Jiang et al., 2024; Karimireddy et al., 2019a). Those methods are designed for full-precision parameters and optimizer states which suffer from high memory consumption.

Another aspect of memory consumption during training is the use of *high-dimensional full-precision optimizer states* which are required for standard optimization algorithms such as Adam (Kingma & Ba, 2014) and momentum-based methods (e.g., heavy-ball momentum (Polyak, 1964) and Nesterov momentum (Nesterov, 1983; 2013)), leading to substantial memory overhead during training. While several approaches aim to reduce this cost by exploiting the tensorial structure of gradients to lower the dimensionality of optimizer states (Zhao et al., 2024; Zmushko et al., 2025; Lialin et al., 2024; Luo et al., 2024; Zhang et al., 2025; He et al., 2025; Zhu et al., 2025), a more direct and general strategy is to quantize these states into low-bit representations, independent of their structure. Although previous works have explored 4-bit quantization of optimizer states (Li et al., 2023; Modoranu et al., 2024; Wang et al., 2024; Alistarh et al., 2017), achieving extremely low bit-width representations, such as ternary quantization, remains largely unresolved. The difficulty lies in the reliance of momentum-based methods on averaging techniques that inherently assume continuous-valued states, making them dependent on floating-point storage and computation. A naïve solution is to re-quantize the momentum states after each update, but this often causes severe precision loss under strict bit budgets. The problem becomes even more acute when optimizing quantized parameters: since each update is already coarse, degraded momentum estimates can lead to poor update decisions. In such cases, the accumulation of inaccuracies from both quantized parameters and quantized optimizer states can produce unstable trajectories. Consequently, repeated quantization (whether stochastic or deterministic) may yield updates that are either excessively large or negligibly small, destabilizing or harming the convergence of the optimization process. This fundamental mismatch between the continuous nature of conventional optimizers and the discrete structure of quantized training represents a key bottleneck to achieving truly memory-efficient optimization.

Additionally, when aiming to apply standard optimization methods which are designed for high-precision gradients, with only extremely limited gradient information, it is unclear how they should be adapted to the coarse gradients, how their convergence guarantees extend, and what information their auxiliary states can meaningfully capture. While sign-based optimization methods have recently attracted attention, their integration with conventional optimizers under strict memory constraints on the parameters and optimizer states remains unexplored.

In this work, we address the several memory sources of the optimization. The first is the parameter domain, where we restrict the parameters to a small predefined set of discrete values, $\theta \in \{r_{\min}, ..., r_{\max}\}^d$, where $d$ is the model dimension, with particular emphasis on ternary parameters, $r_{\min} = -1, r_{\max} = 1$. The second is the gradient information, which we assume that only ternary gradient information is available, i.e., $\tilde{g} \in \{-1, 0, 1\}^d$. The last is the optimizer states, where we focus specifically on the ternary states with $m \in \{-1, 0, 1\}^d$. This extreme regime poses new challenges for training quantized models. Despite the restrictions, the described setting is increasingly relevant for training under tight memory and computational budgets, where every bit of storage matters. To address this gap, we introduce a tailored optimization method that harnesses low-precision gradients to perform momentum-like updates, making quantized optimization viable even under such conditions.

The contributions of our work can be listed as follows:

**Stochastic Ternary Momentum.** We introduce stochastic ternary momentum, an optimization method which includes novel modification of first-order momentum methods that enables the optimizer state and updates to be represented entirely with ternary values. The key innovation is to replace deterministic momentum and parameter updates with a random update rule. At each iteration, we sample Bernoulli masks that determine which coordinates of the momentum and parameters are updated. Instead of applying full-precision magnitudes of the updates, they are realized through probabilistic masks that respect ternary constraints with updates that are limited to $+1$, $-1$ or $0$. Our method builds on the availability of ternary gradient approximations, such as sparse zero-sign representations (Zhang et al., 2024; Chen et al., 2024; Zhou et al., 2025; Bar & Giryes, 2025), discrete non-backward approximations (Di Castro et al., 2024), and the seminal TernGrad approach (Wen et al., 2017).

**Expected Updates Match Deterministic Momentum.** Despite relying on discrete updates, our design is principled: we show that the expected updates of both momentum and parameters exactly match their deterministic counterparts when relying on ternary gradient information. This guarantees that the stochasticity introduced for quantization compatibility does not bias the optimization process, while still enabling convergence with low-memory and fully quantized updates.

**Quantization-Preserving Optimization.** Beyond extreme memory efficiency, a major advantage of our method is that it natively preserves quantization. Unlike conventional approaches, it does not require mixing full-precision and quantized variables, thereby eliminating the repeated overhead of re-quantizing parameters or optimizer states. This property ensures both consistent quantization throughout training and robustness under tight bit budgets.

**Theoretical Convergence Guarantees.** To further motivate our ternary momentum optimization algorithm, we establish convergence guarantees for our method assuming deterministic gradient sign access in two different settings. (i) In the case of separable convex functions over a discrete and bounded domain, our algorithm converges to the global optimum. (ii) For minimizing more general smooth non-convex functions over the unbounded integer set, we provide a convergence guarantee to a local minimum, i.e., a critical point.

**Experimental Results.** We empirically validate our method across diverse architectures and tasks, including image classification on ImageNette and language model fine-tuning with LoRA. Our experiments span ternary parameters, various gradient approximation techniques, and datasets such as IMDB, SST2, COPA, and WinoGrande, using fully connected networks, ViTs, OPT-1.3b and Llama2-7b. Our results demonstrate a reduction in optimizer state memory consumption by a factor of $\log_2(3)/32$ which is equivalent to $\approx 5\%$ compared to the full-precision first-momentum counterpart. Our evaluation demonstrates that our algorithm is both effective and robust, even under extreme memory constraints. These findings highlight the practicality of our approach and open the door to fully quantized optimization in low-resource training environments.

## 2 Related Work

**Quantization.** Neural network quantization is primarily used for inference, where post-training quantization (PTQ) methods dominate. In these approaches, models are first trained in full precision using large computational resources and are then quantized and deployed on resource-limited devices (Jacob et al., 2018; Xiao et al., 2023; Frantar et al., 2023; Han et al., 2016; Liu et al., 2021; Wang et al., 2019b; Li et al., 2016). An alternative paradigm is quantization-aware-training (QAT), where parameters are repeatedly quantized during training to better simulate inference-time behavior. However, QAT methods typically assume access to full-precision gradients and rely on optimizers designed for full-precision updates, which therefore require maintaining high-precision optimizer states in memory (Esser et al., 2020; Zhou et al., 2016; Rastegari et al., 2016; Choi et al., 2018; Courbariaux et al., 2015; Dettmers et al., 2022; Wu et al., 2018).

**Memory Efficient Optimizers.** Recent work has also explored reducing the memory footprint of optimization algorithms. One line of research designs *low-dimensional* optimizers, where gradients and optimizer states are stored or updated in a low-rank or sparse subspace while maintaining full-precision representations (Zhao et al., 2024; Zmushko et al., 2025; Lialin et al., 2024; Luo et al., 2024; Zhang et al., 2025; He et al., 2025; Zhu et al., 2025; Hao et al., 2024; Liang et al., 2024; Yao et al., 2022; Lin et al., 2018; Stich et al., 2018; Alistarh et al., 2018; Wangni et al., 2018). Another line of work focuses on reducing the number of trainable parameters rather than the optimizer states, a common strategy in fine-tuning. Parameter-Efficient Fine-Tuning (PEFT) methods (Han et al., 2024a; Houlsby et al., 2019; Ben-Zaken et al., 2022; Hu et al., 2022), among which LoRA (Hu et al., 2022) is particularly influential (see Han et al. (2024b) for a comprehensive survey), introduce lightweight low-rank adapters or other compact modules into pretrained models to minimize memory usage during fine-tuning. Although PEFT methods do not directly address optimizer quantization, they naturally result in lower memory usage for gradients and optimizer states due to the reduced number of trainable parameters.

The works most relevant to ours are those that explicitly quantize the optimizer states, including methods designed for Adam (Zhang et al., 2025; Modoranu et al., 2024) and other optimizers (Wang et al., 2024; Li

et al., 2023). However, these approaches typically assume access to full-precision gradients and often rely on relatively high bitwidths (e.g., 4 bits), which limits their applicability in highly resource-constrained training settings.

(Di Castro et al., 2024; Bar & Giryes, 2025; Zhou et al., 2025).

# 3 Background and Problem Setup

In the following section we list the relevant background for our ternary momentum algorithm.

**Full-Precision Optimization with Momentum.** We begin by recalling the standard optimization setup involving gradient access and momentum. The objective is to minimize a differentiable loss function $\mathcal{L} : \mathbb{R}^d \to \mathbb{R}$. Starting from initial parameters $\theta_0 \in \mathbb{R}^d$, at each iteration $t = 1, \ldots, T$, we obtain a noisy gradient estimate $g_t \in \mathbb{R}^d$ of the true gradient $\nabla \mathcal{L}(\theta_{t-1})$ and determine the next iterate $\theta_t$ based on the previously observed (noisy) gradients. Typically, this noisy estimate is computed using a randomly sampled mini-batch of training data.

To mitigate the stochastic noise in this process, the heavy-ball method (Polyak, 1964) maintains a running average of past gradients, $m_t = \beta m_{t-1} + (1 - \beta)g_t$, with $m_0 = 0$ and a momentum parameter $\beta \in (0, 1)$. These updates require high-dimensional real optimizer states that are stored in floating-point memory. The parameter update rule is then given by $\theta_t = \theta_{t-1} - \eta_t m_t$, where $\eta_t$ denotes the learning rate at step $t$.

**Quantized Setting.** In the quantized optimization setting, the parameter space is restricted to discrete values. Specifically, we assume $\theta_t \in \mathbb{Z}^d$, with each coordinate bounded between some integers $r_{\min}$ and $r_{\max}$. Throughout this work, we mainly focus on the ternary case, where $r_{\min} = -1$ and $r_{\max} = 1$, so that $\theta_t \in \{-1, 0, 1\}^d$. In addition, we assume access to a quantized gradient information oracle, denoted by $\mathrm{TernGrad}(\theta)$, which given the current parameters $\theta_t$ returns a ternary gradient approximation $\tilde{g} \in \{-1, 0, 1\}^d$. The optimization objective remains the same—to minimize the loss $\mathcal{L}(\theta)$—but under the strict constraint that the parameters must remain within the discrete and bounded range at all times.

In this work, we propose an optimization scheme that employs ternary momentum, offering high memory efficiency while preserving optimization effectiveness. We consider the setting where the momentum is restricted to discrete ternary values, $m_t \in \{-1, 0, 1\}^d$, and depends solely on ternary gradient information, $\tilde{g} \in \{-1, 0, 1\}^d$. Furthermore, our objective is to learn discrete model parameters, $\theta_t \in [r_{\min}, r_{\max}] \subset \mathbb{Z}$, that remain discrete throughout the optimization process.

## 3.1 Naive Quantized Optimization with Ternary Gradients

**Optimization without momentum.** A straightforward approach to quantized optimization with ternary gradient information is to update the parameters directly using only the ternary gradient, $\theta_t = \theta_{t-1} - \tilde{g}_t$. However, such updates can be overly aggressive, often causing instability in the optimization process Bernstein et al. (2018); Karimireddy et al. (2019b). A natural remedy is to scale the update by a learning rate, $\theta_t = \theta_{t-1} - \eta_t \tilde{g}_t$ but this introduces a weakness: it requires mixed-precision operations between the quantized $\theta_t$ and the floating-point $\eta_t \tilde{g}_t \in \mathbb{R}^d$. Hence, the parameters $\theta_t$ must be repeatedly quantized after each step, reintroducing instability and computational overhead Courbariaux et al. (2016).

**Optimization with momentum.** Another natural attempt to stabilize training is to incorporate momentum into the ternary gradient approximation. In this case, the moving average produces real-valued momentum states, $m_t = \beta m_{t-1} + (1 - \beta)\tilde{g}_t \in \mathbb{R}^d$, which in turn yield real-valued parameter updates, $\theta_t = \theta_{t-1} - \eta_t m_t \in \mathbb{R}^d$. To enforce quantization, both the parameters and optimizer states must be re-quantized after every update. Yet, under the aggressive quantization regime considered in this work, repeated re-quantization amplifies the mismatch between the continuous dynamics and their quantized counterparts, leading to updates that are either vanishingly small or excessively large, depending on the hyperparameters. This ultimately undermines stability and prevents reliable convergence.

---

**Algorithm 1** Ternary First Moment

---

**Input:** $\theta_0 \in \{r_{\min}, \ldots, r_{\max}\}^d$ with integers $r_{\min} < r_{\max}$, learning rate $\{\eta_t\}_{t=1}^T$, momentum coefficient $\beta$, TernGrad ternary gradient algorithm, and bounds $r_{\min}, r_{\max}$.
**Initialize:** $m_0 = 0$

**for** $t = 1, \ldots, T$ **do**
    Compute ternary gradient approximation: $\tilde{g}_t = \text{TernGrad}(\theta_{t-1})$
    **Update momentum:**
        Sample vector $e_t^{(m)} \sim \text{Bernoulli}(\beta)$ i.i.d. element-wise where $e_t^{(m)} \in \{0,1\}^d$.
        Momentum Update: $m_t = e_t^{(m)} \odot m_{t-1} + (1 - e_t^{(m)}) \odot \tilde{g}_t$
    **Update parameters:**
        Sample vector $e_t^{(\theta)} \sim \text{Bernoulli}(\eta_t)$ i.i.d. element-wise where $e_t^{(\theta)} \in \{0,1\}^d$.
        Update rule: $\theta_t = \text{clamp}_{[r_{\min}, r_{\max}]} \{\theta_{t-1} - e_t^{(\theta)} \odot m_t\}$
**end for**
Return optimized parameters $\theta_T$

---

## 4 Optimization with Ternary Gradient and Momentum

In this section, we present our momentum-based optimization method designed specifically for the setting of quantized parameters, ternary gradients, and quantized first-order momentum. The method is detailed in Algorithm 1.

The required inputs to our algorithm are quantized initial parameters that are bounded by $r_{\min}$ and $r_{\max}$, learning rate $\eta_t$ that can be adapted over time, a momentum coefficient, and a function for acquiring the ternary approximation of the gradient. At each iteration $t$, we get the ternary information of the gradient using the TernGrad function. Then, unlike standard momentum updates that involve floating-point arithmetic (e.g. $m_t = \beta m_{t-1} + (1 - \beta) g_t$), our approach employs a ternary Bernoulli approximation of the momentum. Specifically, for each entry, we sample a binary random variable $e_t^{(m)} \in \{0, 1\}$ from a Bernoulli distribution with parameter $\beta$. This forms a vector $e_t^{(m)} \in \{0, 1\}^d$, which then dictates element-wise the momentum update: if $e_{t,i}^{(m)} = 1$, the $i$-th component of the momentum $m_{t,i}$ retains its previous value $m_{t-1,i}$; otherwise, if $e_{t,i}^{(m)} = 0$, it is replaced by the corresponding component of the current ternary gradient $\tilde{g}_{t,i}$. This is formally expressed as:

$$m_t = e_t^{(m)} \odot m_{t-1} + (1 - e_t^{(m)}) \odot \tilde{g}_t, \quad e_t^{(m)} \sim \text{Bernoulli}(\beta). \tag{1}$$

The $\beta$ hyperparameter controls the probability of which we rely on past approximations that are stored in the momentum, for higher values of $\beta$ the updates rely more heavily on the history. This stochastic design is critical because it intrinsically ensures that the momentum $m_t$ remains confined to ternary values, thus avoiding the need for high-precision storage and mitigating memory demands.

Next, we describe the parameter update step. Instead of directly applying a floating-point learning rate (e.g., $\theta_t = \theta_{t-1} - \eta_t m_t$), we again leverage a Bernoulli approximation. For each parameter $\theta_{t,i}$, a binary random variable $e_{t,i}^{(\theta)} \in \{0, 1\}$ is sampled from a Bernoulli distribution with parameter $\eta_t$ (the learning rate). This generates a vector $e_t^{(\theta)} \in \{0, 1\}^d$. The parameter update then becomes $\theta_t = \theta_{t-1} - e_t^{(\theta)} \odot m_t$. This effectively means that each parameter is updated by its corresponding momentum component only with a probability equal to the learning rate $\eta_t$.

**Unbiased Updates Property.** Our stochastic update rules of the momentum and parameters are motivated by their expected behavior. Despite the use of integer values, the stochastic nature of the updates maintains consistency with full-precision optimization. In expectation, the updates for both the momentum and the parameters precisely match their full-precision counterparts. Formally,

$$\mathbb{E}[m_t | m_{t-1}, \tilde{g}_t] = \beta m_{t-1} + (1 - \beta) \tilde{g}_t, \quad \text{and} \quad \mathbb{E}[\theta_t | \theta_{t-1}, m_t] = \theta_{t-1} - \eta_t m_t.$$

This theoretical equivalence is crucial as it suggests that the algorithm may achieve similar convergence properties to traditional methods with access to ternary gradients while operating under severe memory and precision constraints.

Our proposed ternary momentum algorithm enjoys several significant strengths, particularly for constrained environments. By storing momentum states in ternary form (requiring $\log_2(3)$ bits) rather than full-precision floats (32 bits), the memory footprint is reduced by a factor of $\frac{\log_2(3)}{32} \approx 0.048$. Moreover, our approach maintains integer-valued parameters $\theta$ and momentum updates throughout training, eliminating the need for mixed-precision operations and avoiding the overhead and potential precision loss of repeated quantization. Updates are performed using randomized masks $e^{(m)}$ and $e^{(\theta)}$ with integer arithmetic, ensuring fully integer-only optimization.

In addition, the expected number of parameter updates can be controlled, providing a proxy for energy efficiency on low-precision hardware. Specifically, for momentum coefficient $\beta$ and learning rate $\eta_t$, the expected number of optimizer state changes per iteration is approximately $(1 - \beta)d$, where $d$ is the number of parameters. Smaller $\eta_t$ or larger $\beta$ reduce the variability of updates, allowing principled regulation of both update sparsity and energy consumption (Stutz et al., 2021).

## 5 Theoretical Analysis

Next, we present a theoretical convergence analysis of our ternary first-moment algorithm (Algorithm 1) under different assumptions on the loss function and parameter domain. Throughout, we assume that the loss function satisfies the following element-wise smoothness assumption.

**Definition 1** (Element-wise smoothness)**.** *A differentiable function $f : \mathbb{R}^d \to \mathbb{R}$ is element-wise $\boldsymbol{L} = (L_1, \ldots, L_d)$-smooth for $L_1, \ldots, L_d \geq 0$, if for any $x, y \in \mathbb{R}^d$,*

$$|f(y) - (f(x) + \nabla f(x) \cdot (y - x))| \leq \frac{1}{2} \sum_{i=1}^{d} L_i (y_i - x_i)^2.$$

This assumption, previously used in the analysis of SignSGD (Bernstein et al., 2018), naturally captures heterogeneous curvature across coordinates and is well suited to sign-based methods, where each coordinate update has a fixed magnitude.

In addition to the smoothness assumption, we distinguish between two settings:

1. **Separable, convex, and smooth setting.** We assume that the objective is separable, i.e., of the form $\mathcal{L}(\theta) = \sum_{i=1}^{d} \mathcal{L}_i(\theta_i)$, where each $\mathcal{L}_i : \mathbb{R} \to \mathbb{R}$ is convex and $L_i$-smooth for $i = 1, \ldots, d$. This is equivalent to assuming that the separable function $\mathcal{L}(\theta)$ itself is convex and $\boldsymbol{L}$-smooth, where $\boldsymbol{L} = (L_1, \ldots, L_d)$. In this setting, convergence is measured in terms of loss suboptimality, $\mathcal{L}(\theta) - \min_{\theta'} \mathcal{L}(\theta')$.

2. **Non-convex and smooth setting.** Here we assume that $\mathcal{L} : \mathbb{R}^d \to \mathbb{R}$ is element-wise $\boldsymbol{L}$-smooth but not necessarily separable or convex. To accommodate this broader function class, we analyze Algorithm 1 in the case where $r_{\min} = -\infty$ and $r_{\max} = \infty$, i.e., no clipping is applied. As is standard in non-convex optimization, convergence is measured in terms of the gradient norm, which reflects convergence to a critical point.

We focus on these two settings, rather than a more general convex or non-convex formulation, due to the intrinsic limitations of sign-based methods with quantized updates of the form $\theta_t - \theta_{t-1} \in \{-1, 0, 1\}^d$. As demonstrated by Karimireddy et al. (2019b), even the simpler SignSGD method fails to converge in certain worst-case instances for more general classes of such problems. While avoiding clipping departs from strict memory efficiency and separability is unlikely to hold in practical scenarios, convergence guarantees established in the above settings provide theoretical support and a foundation for more general analyses.

Finally, for ternary gradient information, we use the element-wise deterministic sign operator over the deterministic gradient at time $t$,

$$\tilde{g}_t = \text{sign}(\nabla\mathcal{L}(\theta_t)) = \begin{cases} +1 & \text{if } \nabla\mathcal{L}(\theta_t) > 0; \\ 0 & \text{if } \nabla\mathcal{L}(\theta_t) = 0; \\ -1 & \text{if } \nabla\mathcal{L}(\theta_t) < 0. \end{cases}$$

We focus on the deterministic setting for simplicity. In stochastic settings, a common strategy to mitigate gradient noise is to employ a very large batch size (Bernstein et al., 2018), effectively reducing the problem to a (semi-)deterministic problem. A possible alternative is to adopt an average-case framework, such as random least-squares problems, which can provide a more refined characterization of gradient noise (Paquette et al., 2021; Paquette & Paquette, 2021). This promising direction lies outside the scope of our work.

In Appendix D, we present synthetic linear regression simulations to illustrate the behavior of our method in both separable and non-separable settings under gradient noise induced by mini-batching. These simulations demonstrate that full-batch training yields the best performance, and that our method remains effective with moderately sized mini-batches, though gradient noise can have a large effect on the final performance for small mini-batches, highlighting the importance of stable ternary gradient estimation.

### 5.1 Separable, Convex and Smooth Objectives

Following, we formally detail the high-probability suboptimality guarantee for Algorithm 1 given a separable, convex and element-wise smooth objective.

**Theorem 1.** *Let $\delta \in (0,1)$, $\beta \in (0,1)$, integers $r_{\min} < r_{\max}$, a separable, convex, and element-wise $\boldsymbol{L}$-smooth function $\mathcal{L}(\theta) = \sum_{i=1}^{d} \mathcal{L}_i(\theta_i)$, and $\theta_0 \in \{r_{\min}, \ldots, r_{\max}\}^d$. For any $T \geq r_{\max} - r_{\min} + \Delta_\beta$, where $\Delta_\beta \triangleq \lceil \log_\beta(\delta/(dT)) \rceil = \lceil \log(dT/\delta)/\log(1/\beta) \rceil$, consider the output $\theta_T$ produced by Algorithm 1 with parameters $\theta_0, r_{\min}, r_{\max}, \beta$, $\eta_t = 1$ for $t = 1, \ldots, T$, and the deterministic signed gradient as* TernGrad. *Then with probability at least $1 - \delta$,*

$$\mathcal{L}(\theta_T) - \min_{\theta' \in [r_{\min}, r_{\max}]^d} \mathcal{L}(\theta') \leq \frac{1}{2} \|\boldsymbol{L}\|_1 \lceil \log(dT/\delta)/\log(1/\beta) \rceil^2,$$

*where the probability is with respect to the random masks $e_t^{(m)}$ for $t = 1, \ldots, T$.*

For a sufficiently large number of steps, Theorem 1 establishes a convergence rate of $\widetilde{O}(\|\boldsymbol{L}\|_1)$. Up to logarithmic factors, this rate is essentially tight: in the worst case, the distance from the minimizer along each coordinate is lower bounded by half the update magnitude, resulting in a worst-case suboptimality of $\Omega(\|\boldsymbol{L}\|_1)$. For example, this lower bound is attained for a simple quadratic objective $\sum_{i=1}^{d} L_i(x_i - 0.5)^2$.

The proof of Theorem 1, provided in Appendix C.1, relies on two key observations. First, due to separability, the gradient with respect to each coordinate always points toward the minimizer. Second, with high probability, the momentum term cannot drive the iterate away from the minimizer for more than a logarithmic number of steps; afterward, the coordinate necessarily returns toward the minimizer.

### 5.2 Non-Convex Smooth Objectives

The result in Section 5.1 addresses a relatively restrictive family of objective functions. We now turn to a more general class of non-convex smooth functions. In this broader setting, we necessarily provide a weaker form of guarantee. Specifically: (1) we no longer assume bounded coordinates, that is, $r_{\min} = -\infty$ and $r_{\max} = \infty$, resulting in optimization over $\mathbb{Z}^d$; and (2) the convergence guarantee concerns minimizing the average $\ell_1$-norm of the gradients rather than the loss suboptimality.[1]

**Theorem 2.** *Let $\mathcal{L} : \mathbb{R}^d \to \mathbb{R}$ be an element-wise $\boldsymbol{L}$-smooth function lower bounded by some $\mathcal{L}^\star \in \mathbb{R}$. Let $\theta_0 \in \mathbb{Z}^d$ and consider the sequence $(\theta_t)_{t=0}^{T}$ generated by Algorithm 1 with parameters $\theta_0, r_{\min} = -\infty, r_{\max} =$*

---

[1]Convergence in non-convex optimization is commonly expressed in terms of minimizing the gradient norm, which corresponds to finding an approximate critical point (Ghadimi & Lan, 2013; Arjevani et al., 2022).

$\infty, \beta \in (0,1)$, $\eta_t = 1$ for $t = 1, \ldots, T$, and the deterministic signed gradient as TernGrad. *Then it holds that*

$$\frac{1}{T} \sum_{t=1}^{T} \mathbb{E}[\|\nabla \mathcal{L}(\theta_{t-1})\|_1] \leq \frac{\mathcal{L}(\theta_0) - \mathcal{L}^\star}{(1-\beta)T} + \frac{(1+5\beta)\|\boldsymbol{L}\|_1}{2(1-\beta)^2}.$$

Similar to Theorem 1, the result above guarantees convergence to a neighborhood of size $O(\|\boldsymbol{L}\|_1)$, which is unavoidable due to the fixed magnitude of the updates. On the other hand, Theorem 2 complements Theorem 1 by showing that the algorithm converges, in terms of the average gradient norm, for a broader class of objectives. Note that while Theorem 2 fixes $\eta_t = 1$ for $t = 1, \ldots, T$, we prove a more general version that supports arbitrary $\eta_1, \ldots, \eta_T$, including the schedules $\eta_t = 1/t$ and $\eta_t = 1/\sqrt{t}$, although these schedules yield a slower convergence rate. See Appendix C.2 for full details.

Taken together, these results help explain the practical success of Algorithm 1 on favorable objectives–either when the optimization landscape exhibits separability, or when convergence to a critical point corresponds to strong empirical performance, as is often implicitly assumed when applying gradient-based methods to non-convex problems.

## 6 Experiments

We evaluate our proposed method across both image and language domains using various models to demonstrate its effectiveness and versatility. We include various datasets, tasks, and ternary gradient approximation methods.

**Datasets and Tasks**

1. **Sentiment Analysis with IMDB:** We perform sentiment analysis using the IMDB dataset, leveraging BERT features (Devlin et al., 2019). The model architecture consists of a fully-connected network with three layers, each having a hidden width of 4096.

2. **Image Classification with ImageNette:** We utilize a subset of ImageNet, known as ImageNette (Howard, 2019), which consists of 10 classes. For this task, we employ the Vision Transformer (ViT) architecture (Dosovitskiy, 2020). We investigate two variants of our method: Hybrid 1 Variant: In this configuration, we replace the feed-forward network in each transformer layer with ternary weights. Hybrid 2 Variant: This variant further involves replacing the key ($K$), query ($Q$), and value ($V$) matrices in the attention block while retaining the prediction head in full-precision. The exact number of ternary parameters used in both variants is detailed in Table 1.

3. **Fine-Tuning of Language Models:** We fine-tune OPT-1.3b (Zhang et al., 2022) and Llama2-7b (Touvron et al., 2023) language models for with SST-2 (Socher et al., 2013), COPA (Wang et al., 2019a) and WinoGrande (Sakaguchi et al., 2021)datasets using LoRA. In this setup, all learned parameters are ternary, while the parameters that remain frozen are maintained in full precision (unless stated otherwise). LoRA which lowers the dimension of learned parameters during fine-tuning is highly compatible with our method's aim to reduce memory during training. Importantly, since our method requires the initial parameters to be discrete fine-tuning all the parameters with our algorithm require some post training quantization. Hence, we use LoRA fine-tuning which introduces new parameters during fine-tuning so we initialize them in a quantized manner.

We use the experimental setup and the code of Di Castro et al. (2024) for IMDB and ImageNette (results in Table 1). Furthermore, we use the code of Zero-Order Optimization Benchmarking framework (Zhang et al., 2024) which include code for fine-tuning LLMs with zero-order optimization method. We specifically used it for LorA fine-tuning (results in Table 3).

**Ternary Gradient Approximations**

Our method requires access to ternary gradient approximations that can potentially be acquired efficiently. However, in our experiments we mostly simulate this approximation. We use different method for ternarizing

the gradient during optimization and we asses the robustness of the benefits of our ternary momentum to different approximation methods. We use the following methods:

1. **Deterministic Ternarization:** In this approach, 10% of the full gradient entries with the smallest magnitude are set to 0, while the remaining entries are assigned values based on their sign. For this approximation we calculated the gradient of the current batch using a backward pass.

2. **Stochastic TernGrad (Wen et al., 2017):** A widely known approach samples the sparsity mask according to the gradient magnitudes, after which a sign operator is applied to the selected non-zero entries. This method is well-known, it was designed for efficient distributed optimization where the batched gradients is computed in full-precision and then quantized to ternary values.

3. **No Backward (Di Castro et al., 2024):** This method does not involve a backward pass and relies on a discrete algorithm to assess the contribution of each parameter. The parameters are then updated using only $\{\pm 1\}$ values for influential parameters, yielding a ternary update step. As a result, the method provides an efficient alternative for gradient estimation. Since this method aims to maintain parameters within a bounded integer domain without FP operations, it does not employ sophisticated optimization schemes. Our method, which is designed for optimization in the same constrained regime, naturally complements their ternary gradient approximation, making it well-suited for evaluation in this setting.

4. **Sparse Zero-Order (ZO) (Zhang et al., 2024):** This method employs a zero-order optimization approach (Liu et al., 2019) to estimate the gradient sign. Specifically, it perturbs the parameters by adding and subtracting small noise and determines the gradient sign based on the corresponding increase or decrease in the loss value. The sparsity arises from zeroing a randomly selected subset of parameters, where 30% of them are non-zero. Full-precision operation are required for the perturbation and then a full-precision forward is needed. The Backward pass is not needed for zero-order approximation method.

Deterministic and TernGrad ternary gradient approximation methods require full access to the gradients and are therefore used primarily as simulations. In contrast, the latter two methods do not rely on gradient information, making them more suitable for the studied use case. Additionally, although the Sparse ZO approximation does not require full gradient access, the parameter perturbations it introduces still necessitate storing a full-precision copy of the model, which increases memory requirements.

**Optimizer Baselines.** Since current optimization schemes rely on high-precision gradient information rather than ternary one and are focused on high-precision updates of the parameters, using off-the-shelf optimizers requires additional adjustments to the setting and hence we leave that for future work. Hence, we compare our method with two optimizers which are very relevant to the scenario we are focused on.

1. **Stochastic Updates without Momentum:** The update steps are chosen according to a randomly sampled masks, but no momentum is used. The update step is $\theta_t = \theta_{t-1} - e_t^{(\theta)} \tilde{g}_t$ where $e_t^{(\theta)} \sim$ Bernoulli$(\eta_t)$ (as in the parameter update step in Algorithm 1). Note that this baseline is memory efficient since it does not require storing 1-bit optimizer states (see Table 2).

2. **Full-Precision Momentum:** The update steps are applied according to full-precision momentum. The momentum updates are $m_t = \beta m_{t-1} + (1-\beta)\tilde{g}_t$ and the parameter updates are performed with stochastic rounding $\theta_t = SR(\theta_{t-1} - e_t^{(\theta)} \tilde{g}_t)$ where $e_t^{(\theta)} \sim$ Bernoulli$(\eta_t)$. Note that the original optimizer relies on full-precision gradients, and its effectiveness under limited gradient information, such as in the case of ternary gradient approximation, remains unexplored. Most importantly, additional full-precision optimizer states are required to be stored.

In all experiments, we use a learning rate of $0.75/t$, which depends on the iteration number, following Di Castro et al. (2024). In Appendix A, we provide additional details on the hyperparameter used and hyperparameter search for the initial learning rate, learning rate scheduler, and momentum coefficient.

| Ternary Approx. | Optimizer | IMDB | ImageNette Hybrid 1 | ImageNette Hybrid 2 |
|---|---|---|---|---|
| | # of parameters | 19.9M | 38.33M | 38.33M |
| | # of Ternary | 19.9M | 25.1M | 37.3M |
| FP model+grad | AdamW | 87.3 | 90.8 | 90.8 |
| Deterministic | No Momentum | $86.75_{\pm0.08}$ | $83.6_{\pm0.43}$ | $78.32_{\pm0.37}$ |
| | FP Momentum | $86.19_{\pm0.08}$ | $86.78_{\pm0.28}$ | $\mathbf{82.89}_{\pm0.13}$ |
| | Ternary Momentum | $\mathbf{86.86}_{\pm0.05}$ | $\mathbf{87.69}_{\pm0.45}$ | $81.44_{\pm0.29}$ |
| Stochastic | No Momentum | $85.27_{\pm0.19}$ | $89.58_{\pm0.36}$ | $85.46_{\pm0.26}$ |
| | FP Momentum | $82.9_{\pm0.36}$ | $89.58_{\pm0.15}$ | $85.29_{\pm0.19}$ |
| | Ternary Momentum | $\mathbf{85.28}_{\pm0.05}$ | $\mathbf{89.62}$ | $\mathbf{86.35}_{\pm0.29}$ |
| No Backward | No Momentum | $83.04_{\pm0.23}$ | $85.1_{\pm0.29}$ | $75.0_{\pm0.17}$ |
| | FP Momentum | $85.29_{\pm0.13}$ | $83.97_{\pm0.37}$ | $71.44_{\pm0.15}$ |
| | Ternary Momentum | $\mathbf{85.57}_{\pm0.15}$ | $\mathbf{88.23}_{\pm0.23}$ | $\mathbf{79.4}_{\pm0.27}$ |

Table 1: Accuracy for training from scratch ternary parameters with multiple ternary gradient approximation methods. The FP baseline refers to standard practice of training full-precision models which utilize AdamW. $\pm$ denotes the standard deviation, and bold indicates the highest average accuracy for each baseline.

| | Theoretical | Fully Connected | Hybrid 1 | Hybrid 2 | OPT-1.3b LoRA | Llama2-7b LoRA |
|---|---|---|---|---|---|---|
| No Momentum | 0 | 0 | 0 | 0 | 0 | 0 |
| FP Momentum | $d$ | 79.6 | 95.75 | 142.2 | 6 | 16.78 |
| Ternary Momentum | $d/\log_2(3)$ | 3.94 | 4.74 | 7.1 | 0.3 | 0.84 |

Table 2: Optimizer states memory in MB.

## 6.1 Results

**Training from scratch.** Our initial experiments on standard classification datasets (Table 1) demonstrate that the proposed ternary first momentum method effectively learns a model from scratch across diverse tasks. We observe that our method, employing all ternary approximation strategies, generally outperforms the quantized baselines, affirming the benefits of incorporating even an extremely low precision momentum for effective optimization.

As expected, all tested configurations exhibit some degradation in accuracy compared to their full-precision training (parameters and optimizer states) counterparts. However, these differences are often small, suggesting that models trained with ternary momentum can achieve competitive performance on standard classification tasks. Specifically, when both parameters and momentum are ternary, we observe an accuracy degradation of up to 2% for IMDB, 3.1% for ImageNette with Hybrid 1, and 9.5% for ImageNette with Hybrid 2. This pattern suggests that the degradation tends to increase as the number of ternary parameters being optimized grows.

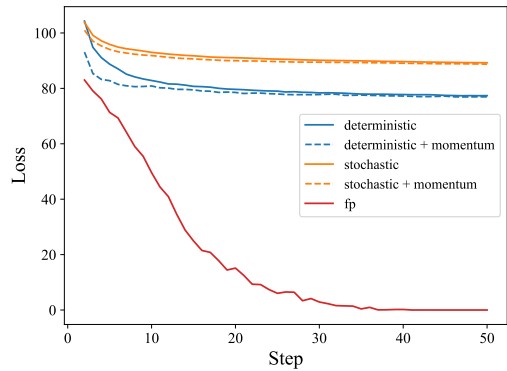

Figure 1: Training loss curves (cross-entropy loss) on the IMDB dataset using deterministic and stochastic ternary gradient approximations. Using momentum results in faster convergence of the training loss. FP line represents the standard practice of training full-precision with AdamW.

Notably, in the ImageNette Hybrid 2 task with deterministic ternary approximation, our ternary momentum method showed some degradation compared to the

| Ternary Approx. | Optimizer | SST2 OPT-1.3b | COPA Llama2-7b | WinoGrande Llama2-7b |
|---|---|---|---|---|
| FP param+grad | AdamW | 96.7 | 85.0 | 69.5 |
| Deterministic | No Momentum | $93.0_{\pm 0.30}$ | 73.0 | $68.6_{\pm 0.26}$ |
| | Ternary Momentum | $\mathbf{93.4}_{\pm 0.17}$ | **76.0** | $\mathbf{70.2}_{\pm 0.26}$ |
| Stochastic | No Momentum | $93.6_{\pm 0.24}$ | 84.0 | $\mathbf{66.6}_{\pm 0.07}$ |
| | Ternary Momentum | $\mathbf{93.7}_{\pm 0.07}$ | **85.0** | $66.1_{\pm 0.14}$ |
| No Backward | No Momentum | $93.2_{\pm 0.39}$ | 80.0 | $68.5_{\pm 0.56}$ |
| | Ternary Momentum | $\mathbf{93.5}_{\pm 0.52}$ | **83.0** | $\mathbf{68.9}_{\pm 0.07}$ |
| Sparse ZO | No Momentum | $\mathbf{93.4}_{\pm 0.41}$ | **84.0** | $67.4_{\pm 0.47}$ |
| | Ternary Momentum | $\mathbf{93.4}_{\pm 0.17}$ | **84.0** | $\mathbf{68.3}_{\pm 0.39}$ |

Table 3: Accuracy results of fine-tuning ternary parameters of LoRA with SST2, COPA, and WinoGrande. FP baseline refers to standard practice for fine-tuning with LoRA which utilizes AdamW. Higher is better and the highest of each baseline is marked in bold. The median accuracy is reported for COPA. $\pm$ denotes the standard deviation.

| Method | IMDB | ImageNette |
|---|---|---|
| Ternary | 83.04 | 85.46 |
| Ternary+Mom. | 85.57 | 86.35 |
| Ternary+Quant. Act. | 79.86 | 39.42 |
| Ternary+Mom.+Quant. Act. | 79.92 | 42.67 |

| Method | SST2 |
|---|---|
| Ternary | 93.58 |
| Ternary+Momentum | 93.69 |
| Quantized Weights | 93.12 |
| Quantized Weights+Momentum | 93.27 |

Table 4: Comparison of accuracy results with quantized activation to 8 bits. Our ternary momentum enjoys accuracy benefits which suggests it can be used with activation quantization for memory reduction.

Table 5: Effect of pretrained parameters quantization (to 8 bits) and ternary momentum on SST2 fine-tuning accuracy. The results suggests that ternary momentum can be used with quantized pretrained parameters.

full-precision baseline, likely due to higher sensitivity to gradient quantization noise in this larger and deeper model. Nevertheless, it still outperformed the no-momentum baseline, highlighting its robustness and strong memory–efficiency trade-off as detailed below. For other baselines, including ImageNette Hybrid-1 and IMDB with stochastic ternary gradient approximation, the observed performance gains are more modest. This may be attributed to suboptimal hyperparameter choices or to the increased noise introduced by the stochastic nature of the gradient estimates.

Furthermore, Fig. 1 demonstrates that momentum effectively accelerates loss convergence, showcasing an additional benefit of its use. The observed gap between full-precision and ternary optimization is expected, as quantization introduces an inherent approximation error in both the parameters and the optimization trajectory. While the approximation error introduced by quantized models is unavoidable, it remains unclear whether improved algorithms can mitigate the degradation in convergence caused by quantized optimization.

**Fine-Tuning LLMs with LoRA.** The results for fine-tuning LLMs with LoRA (Table 3) further underscore the practical utility of our ternary momentum method. We observe that our approach generally achieves superior performance compared to optimizing ternary models with ternary gradients without momentum across various downstream tasks. This demonstrates that the stochastically approximated momentum effectively enhances optimization stability and performance, even when applied to large, pre-trained models.

Our method's inherent memory efficiency complements LoRA, which reduces the number of trainable parameters. By minimizing the optimizer state footprint, the ternary first-momentum method enables fine-tuning of larger models under tight memory constraints. Combined with ternary gradients, it achieves predominantly ternary memory usage for both gradients and momentum, yielding substantial overall memory savings during training.

Notably, we observed a minor performance degradation specifically for the WinoGrande dataset when using a stochastic ternary approximation. We hypothesize this particular instance of degradation may be attributed to the inherent noise and adversarial nature of the WinoGrande dataset interacting unfavorably with the stochastic momentum. For other baselines, specifically SST2 and COPA with Sparse-ZO, the more limited improvements may be explained by the aggressive stochasticity inherent in the gradient approximation scheme.

**Memory of optimizer states.** Table 2 comprehensively details the memory required for optimizer states across various models. The results clearly demonstrate the drastic reduction of memory footprint compared to full-precision momentum optimizers. Specifically, our method is using only $\log_2(3)/32 \approx 5\%$ for the first momentum buffer compared to a full-precision momentum. While naturally requiring more memory than an optimizer completely devoid of momentum, the ternary momentum memory cost is demonstrably minimal for the performance gains it provides, presenting an optimized trade-off. This magnitude of memory savings becomes particularly critical for very large models, where memory constraints are often the primary training bottleneck.

**Ternary momentum with other memory reduction methods.** In Tables 4 and 5 we report additional memory reduction experiments, focusing on activation quantization and weight quantization of the pre-trained parameters. Across all tested settings, incorporating ternary momentum consistently improves accuracy. The effect is particularly pronounced on ImageNette, where aggressive memory reduction leads to substantial performance degradation, and the proposed method mitigates this loss. The results suggest that our method can be used as a complementary approach for reducing memory consumption during training, alongside other quantization methods.

These findings demonstrate the ternary momentum method's position as a highly effective and memory-efficient optimization strategy for large-scale machine learning, particularly beneficial in resource-constrained environments.

## 7 Conclusion, Limitation, and Future Work

In this work, we introduce a ternary first-momentum optimization algorithm, provide theoretical motivation for its design, and demonstrate its empirical potential. Our results show that optimizer states can retain meaningful information even under an extremely low bit-width setting with ternary quantization.

While the proposed method demonstrates promising results, its practical applicability is currently limited by the lack of efficient and accurate ternary gradient approximation mechanisms that operate in a fully quantized manner. Furthermore, we do not account for the practical overhead typically associated with using ternary values on current hardware. To deploy our method on real edge devices or chips, the gradient sign approximation must be fully quantized, and additional hardware-specific adjustments are required to enable forward (and, if necessary, backward) passes that align with the device's capabilities. Moreover, the model architecture, or context length, may need to be chosen carefully to satisfy memory constraints. Although our method shows benefits in several settings, it is not intended as a replacement for full-precision industrial-scale pre-training, which is beyond the scope of this work.

This work opens several directions for future research. One promising avenue is extending the approach to stochastic and quantized variants of second-moment optimizers like Adam (Kingma & Ba, 2014). The algorithmic changes required for second moment with ternary gradient approximation are non-trivial due to the nature of powers of ones, i.e. $(-1)^2 = 1^2 = 1$ which limits the ability to weight parameters differently. Additionally, combining low bit-width quantization with low-dimensional optimization techniques could further enhance memory and computational efficiency. Another interesting direction is to explore adaptive masking strategies, where the update mask is sampled based on a dedicated saliency score per parameter or adjusted over time (see Paquette & Paquette (2021)) rather than being uniformly applied across parameters.

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

## A    Experiment Details

| **ImageNette ViT** | |
| --- | --- |
| Batch Size | 200 |
| FP LR | $6 \times 10^{-4}$ |
| WD | 0.05 |
| Iterations | 5000 |
| Embed. dim | 512 |
| Layers | 12 |
| Heads | 16 |
| Init. density | 0.9 |

| **IMDB Fully-Connected** | |
| --- | --- |
| Batch Size | 256 |
| Epochs | 50 |
| Layers | 3 |
| Input dim | 768 |
| Hidden dim | 4096 |
| Init. density | 0.9 |

| **LM Fine-Tuning w. LoRA** | |
| --- | --- |
| Batch size | 16 |
| Iterations | 20000 |
| WD | 0 |
| LoRA $\alpha$ | 16 |
| LoRA $r$ | 8 |
| Init. density | 0.9 |

Figure 2: Hyperparameters used with different settings.

**Hyperparameters Search.** Results of grid search with multiple values of momentum coefficient $\beta$ and initial learning rate $\eta_0$. The results show the benefits of the chosen hyperparameters.

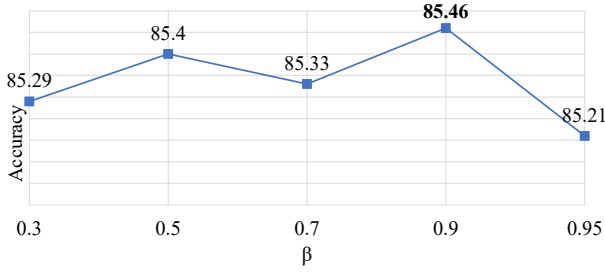

Figure 3: Momentum coefficient, $\beta$, hyperparameter search.

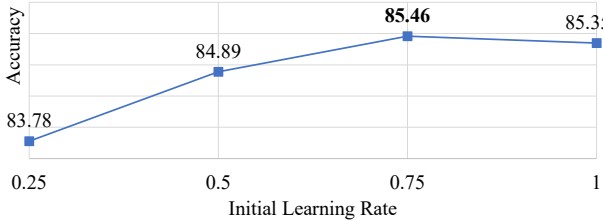

Figure 4: Initial learning rate search.

In Table 6 and Fig. 5, we compare the inverse-time learning rate scheduler with the constant schedule $1/\sqrt{t}$.

| $\eta$ | $\eta/\sqrt{t}$ | $\eta/t$ |
|--------|-----------------|----------|
| 80.79  | 80.95           | 85.57    |

Table 6: Accuracy results with different learning rate schedulers with IMDB and No Backward sign approximation. For constant $\eta$ we use $\eta = 0.1$ for all other scheduler we use $\eta = 0.75$.

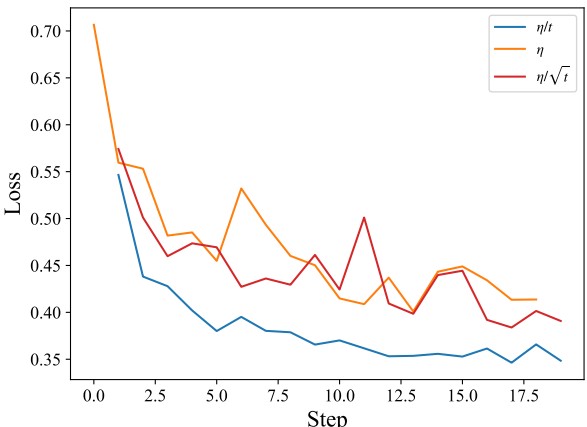

Figure 5: Cross entropy test loss with different learning rate schedulers.

## B  Additional Results

Additional comparisons with the use of our ternary momentum.

|                                | FP AdamW | No Backward – | No Backward FP Momentum | No Backward Ternary Momentum |
|--------------------------------|----------|---------------|-------------------------|------------------------------|
| Optimization parameters [#M]   | 38.9     | 0             | 19.9                    | 19.9                         |
| Changes [#B]                   | 585.06   | 0.19          | 195.21                  | 19.69                        |
| Energy [J]                     | 2.851    | 0.936         | 1.084                   | 1.010                        |

Table 7: We report the number of parameter changes summed with the optimizer-state changes, together with the corresponding expected energy consumption. For our method with momentum, the reported number of changes constitute an upper bound, since updates are driven by the momentum term and the exact number of zero updates cannot be directly approximated. In contrast, without momentum, the expected number of changes is explicitly controlled through deterministic masking, while momentum can be interpreted as an additional masking mechanism applied on top of this deterministic scheme. The approximated difference in energy consumption when using ternary momentum is small, while accuracy improves at the cost of a modest additional energy overhead.

## C  Proofs of Section 5

### C.1  Proof of Theorem 1

Note that for all $t \in [T]$, since $\eta_t = 1$, it holds that $e_t^{(\theta)} = \mathbf{1}$. Hence,

$$\theta_{t,i} = \theta_{t-1,i} - m_{t,i}, \quad \text{and} \quad |\theta_{t,i} - \theta_{t-1,i}| \leq 1.$$

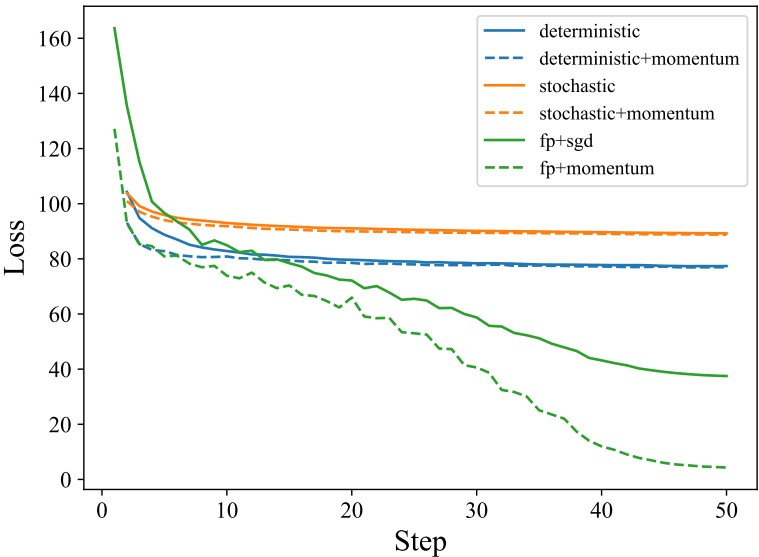

Figure 6: Training loss curves (cross-entropy loss) on the IMDB dataset using deterministic and stochastic ternary gradient approximations. FP lines represent full-precision training with SGD, with and without momentum. Momentum accelerates convergence in the ternary case, while in the full-precision case it improves the final training loss but does not noticeably accelerate convergence.

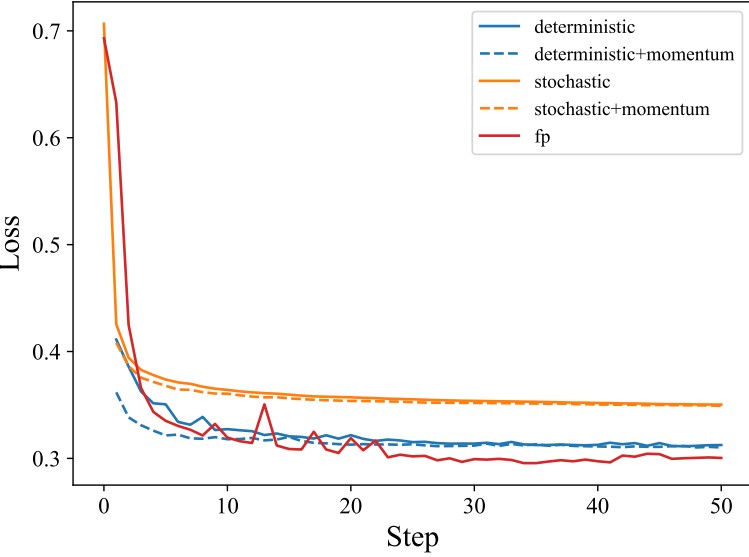

Figure 7: Test loss curves (cross-entropy) on the IMDB dataset using deterministic and stochastic ternary gradient approximations. The FP line represents full-precision training with AdamW. While in Fig. 1 the training loss demonstrates a large gap between FP and ternary optimization, the gap in test loss is smaller, explaining the small accuracy difference reported in Table 1.

| Method | Params | Grads | Optim | Activation | Total |
|---|---|---|---|---|---|
| *IMDB* | | | | | |
| FP | 79.60 | 79.60 | 159.20 | 5.0 | 323.40 |
| Ternary | 3.94 | 3.94 | 0.00 | 5.0 | 12.89 |
| Ternary+Tern. Mom. | 3.94 | 3.94 | 3.94 | 5.0 | 16.83 |
| *ViT* | | | | | |
| FP | 149.20 | 149.20 | 298.40 | 21 620.0 | 22 216.80 |
| Ternary | 7.46 | 7.46 | 0.00 | 21 620.0 | 21 634.92 |
| Ternary+Tern. Mom. | 7.46 | 7.46 | 7.46 | 21 620.0 | 21 642.38 |
| *OPT1-3b* | | | | | |
| FP | 5335.04 | 6.1440 | 12.2880 | 1024.0 | 6377.47 |
| Ternary | 5324.80 | 0.3072 | 0.0000 | 1024.0 | 6349.11 |
| Ternary+Tern. Mom. | 5324.80 | 0.3072 | 0.3072 | 1024.0 | 6349.41 |

Table 8: Memory Footprint Comparison (All values in MB). For IMDB our method show clear benefits, for other settings the benefits are shown in the optimization memory. Those setting are presented mainly to demonstrate the benefits in accuracy, see Tables 1 and 3.

| | Deterministic | Stochastic | No Backward |
|---|---|---|---|
| FP (with Adam) | 87.3 | – | – |
| No Momentum | 86.75 | 85.27 | 83.04 |
| Ternary Momentum (quant. after step) | 76.88 | 85.47$^\star$ | 84.4$^\star$ |
| Ternary Momentum (stochastic, ours) | 86.86 | 85.28 | 85.57 |

Table 9: Performance comparison using non-stochastic updates and momentum on the IMDB dataset. Momentum is updated according to $m_t = \beta m_{t-1} + (1-\beta)\tilde{g}_t$ and the parameters are updated $\theta_t = \theta_{t-1} - \eta_t m_t$ and then both momentum and parameters are quantized to ternary values using stochastic quantization. Results indicate that while accuracy gains are observed in some specific configurations, the benefits of non-stochastic momentum and updates are inconsistent across gradient approximation. Furthermore, the observed training instability ($^\star$ denotes settings where only converged seeds are reported) suggests that additional regularization or optimization measures are required for reliable convergence.

**Step (i): the good event.** Fix $i \in [d]$. We define the following good event:

$$E_i \triangleq \left\{ \forall t \in [T - \Delta_\beta + 1], \exists d \in \{0, 1, \ldots, \Delta_\beta - 1\} \text{ s.t. } e_{t+d,i}^{(m)} = 0 \right\}.$$

Fixing some $t \in [T - \Delta_\beta + 1]$,

$$\Pr(e_t^{(m)} = e_{t+1}^{(m)} = \ldots = e_{t+\Delta_\beta - 1}^{(m)} = 1) = \beta^{\Delta_\beta} \le \delta/(dT),$$

where the last inequality follows from the definition of $\Delta_\beta$. Performing a union bound over at most $T$ windows, $\Pr(E_i) \ge 1 - \delta/d$. Defining $E_G = \cap_{i \in [d]} E_i$ and performing a union bound, $\Pr(E_G) \ge 1 - \delta$.

The rest of the proof will follow deterministically, conditioned on $E_G$.

**Step (ii): distance from the minimizer.** Let $\Theta = [r_{\min}, r_{\max}]$ and let $\Theta_i^\star \triangleq \arg\min_{\theta_i \in \Theta} \mathcal{L}_i(\theta_i)$. For any $t \in [T]$, let $D_i(\theta_t)$ be the distance of $\theta_{t,i}$ to the set of minimizers $\Theta_i^\star$. I.e.,

$$D_i(\theta_t) \triangleq \min_{\theta_i^\star \in \Theta_i^\star} |\theta_{t,i} - \theta_i^\star|.$$

Our goal is to bound $D_i(\theta_t) \le \Delta_\beta$ for all $t \in [\Delta_\beta + r_{\max} - r_{\min}, T]$. Let $\tau \in [t]$ be the maximal iterate with $D_i(\theta_\tau) < 1$. To establish the existence of such $\tau$, first, note that conditioned on $E_i$, there is some $s \le \Delta_\beta$ with

$\tilde{g}_{s,i} = m_{s,i}$. Second, since $\tilde{g}_{s,i} = \text{sign}(\theta_{s-1,i} - \theta_i^\star)$ (for all $\theta_i^\star$) from convexity in 1-d, $m_{s,i} = \text{sign}(\theta_{s-1,i} - \theta_i^\star)$, after which $D_i(\theta_s), D_i(\theta_{s+1}), \ldots$ must decrease by 1 in each step until reaching some $s'$ with $D_i(\theta_{s'}) < 1$ (as all the gradients are aligned with the momentum until the iterates cross the set of minimizers). According to the domain size, it must happen after at most $\Delta_\beta + r_{\max} - r_{\min} < T$ steps.

Now, if $t - \tau < \Delta_\beta$, then $D_i(\theta_t) \le D_i(\theta_\tau) + t - \tau \le \Delta_\beta$. Otherwise, conditioned on $E_i$, there is some $s \in [\tau + 1, \tau + \Delta_\beta]$ with $m_{s,i} = \text{sign}(\theta_{s-1,i} - \theta_i^\star)$ (for any $\theta_i^\star \in \Theta_i^\star$), and $D_i(\theta_s), D_i(\theta_{s+1}), \ldots$ must decrease by 1 in each step until reaching below the 1 threshold, either contradicting the maximality of $\tau$ or enforcing $D_i(\theta_t) \le D_i(\theta_{s-1}) \le \Delta_\beta$. Hence, $D_i(\theta_t) \le \Delta_\beta$.

Hence, we have established bounds for each $i \in [d]$ that hold with probability at least $1 - \delta$ (the probability of $E_G$), such that

$$\mathcal{L}(\theta_T) - \min_{\theta \in \Theta^d} \mathcal{L}(\theta) \le \frac{1}{2} \|\boldsymbol{L}\|_1 \Delta_\beta^2. \qquad \square$$

## C.2 Proof of Theorem 2

Here we prove a more general version of Theorem 2 that supports an arbitrary sequence $\eta_1, \ldots, \eta_T$. Theorem 2 then follows immediately by substituting $\eta_t = 1$ for $t = 1, \ldots, T$.

**Theorem 3.** *Let $\mathcal{L} : \mathbb{R}^d \to \mathbb{R}$ be an element-wise $\boldsymbol{L}$-smooth function lower bounded by some $\mathcal{L}^\star \in \mathbb{R}$. Let $\theta_0 \in \mathbb{Z}^d$ and consider the sequence $(\theta_t)_{t=0}^T$ generated by Algorithm 1 with parameters $\theta_0, r_{\min} = -\infty, r_{\max} = \infty, \beta \in (0, 1)$, some $\eta_1, \ldots, \eta_T > 0$, and the deterministic signed gradient as* TernGrad. *Then it holds that*

$$\frac{1}{\sum_{t=1}^T \eta_t} \sum_{t=1}^T \eta_t \mathbb{E}[\|\nabla \mathcal{L}(\theta_{t-1})\|_1] \le \frac{\mathcal{L}(\theta_0) - \mathcal{L}^\star}{(1 - \beta) \sum_{t=1}^T \eta_t} + \frac{(1 + 5\beta) \|\boldsymbol{L}\|_1}{2(1 - \beta)^2}.$$

In order to prove the theorem, we will use the following lemma. Its proof follows.

**Lemma 1.** *Let $\theta \in \mathbb{R}^d$, $\Delta \in \{-1, 0, 1\}^d$, $\boldsymbol{L} = (L_i)_{i=1}^d \in \mathbb{R}_+^d$, and let $\mathcal{L} : \mathbb{R}^d \to \mathbb{R}$ be a differentiable function satisfying*

$$\forall x, y \in \mathbb{R}^d \quad : \quad |\mathcal{L}(y) - (\mathcal{L}(x) + \nabla \mathcal{L}(x) \cdot (y - x))| \le \frac{1}{2} \sum_{i=1}^d L_i (y_i - x_i)^2.$$

*Then it holds that*

$$\|\nabla \mathcal{L}(x + \Delta) - \nabla \mathcal{L}(x)\|_1 \le 3 \|L\|_1.$$

*Proof.* By the smoothness assumption,

$$\mathcal{L}(\theta_t) - \mathcal{L}(\theta_{t-1}) \le \nabla \mathcal{L}(\theta_{t-1}) \cdot (\theta_t - \theta_{t-1}) + \frac{1}{2} \sum_{i=1}^d L_i (\theta_{t,i} - \theta_{t-1,i})^2$$

$$= -\nabla \mathcal{L}(\theta_{t-1}) \cdot (e_t^{(\theta)} \odot m_t) + \frac{1}{2} \sum_{i=1}^d L_i (e_{t,i}^{(\theta)} m_{t,i})^2.$$

Summing for $t = 1, \ldots, T$, replacing $\mathcal{L}(\theta_T) \ge \mathcal{L}^\star$, and rearranging,

$$\sum_{t=1}^T \mathbb{E}[\nabla \mathcal{L}(\theta_{t-1}) \cdot (e_t^{(\theta)} \odot m_t)] \le \mathcal{L}(\theta_0) - \mathcal{L}^\star + \frac{1}{2} \sum_{t=1}^T \sum_{i=1}^d L_i \mathbb{E}[(e_{t,i}^{(\theta)} m_{t,i})^2].$$

By the law of total expectation, $\mathbb{E}[\nabla \mathcal{L}(\theta_{t-1}) \cdot (e_t^{(\theta)} \odot m_t)] = \eta_t \mathbb{E}[\nabla \mathcal{L}(\theta_{t-1}) \cdot m_t]$, and noting that $e_{t,i}^{(\theta)} m_{t,i} \in \{-1, 0, 1\}$, $\mathbb{E}[(e_{t,i}^{(\theta)} \odot m_{t,i})^2] \le \eta_t$. Hence,

$$\sum_{t=1}^T \eta_t \mathbb{E}[\nabla \mathcal{L}(\theta_{t-1}) \cdot m_t] \le \mathcal{L}(\theta_0) - \mathcal{L}^\star + \frac{\|\boldsymbol{L}\|_1}{2} \sum_{t=1}^T \eta_t.$$

Next we will focus on the $\mathbb{E}[\nabla\mathcal{L}(\theta_{t-1}) \cdot m_t]$.

$$\begin{aligned}
\mathbb{E}[\nabla\mathcal{L}(\theta_{t-1}) \cdot m_t] &= \mathbb{E}[\|\nabla\mathcal{L}(\theta_{t-1})\|_1 + \nabla\mathcal{L}(\theta_{t-1}) \cdot (m_t - \tilde{g}_t)] && (\nabla\mathcal{L}(\theta_{t-1}) \cdot \tilde{g}_t = \|\nabla\mathcal{L}(\theta_{t-1})\|_1) \\
&= \mathbb{E}[\|\nabla\mathcal{L}(\theta_{t-1})\|_1 + \nabla\mathcal{L}(\theta_{t-1}) \cdot (e_t^{(m)} \odot (m_{t-1} - \tilde{g}_t))] \\
&= \mathbb{E}[\|\nabla\mathcal{L}(\theta_{t-1})\|_1 + \beta\nabla\mathcal{L}(\theta_{t-1}) \cdot (m_{t-1} - \tilde{g}_t)] && (\theta_{t-1}, m_{t-1}, \tilde{g}_t \text{ independent from } e_t^{(m)}) \\
&= \mathbb{E}[(1-\beta)\|\nabla\mathcal{L}(\theta_{t-1})\|_1 + \beta\nabla\mathcal{L}(\theta_{t-1}) \cdot m_{t-1}] && (\nabla\mathcal{L}(\theta_{t-1}) \cdot \tilde{g}_t = \|\nabla\mathcal{L}(\theta_{t-1})\|_1) \\
&= \mathbb{E}[(1-\beta)\|\nabla\mathcal{L}(\theta_{t-1})\|_1 + \beta\nabla\mathcal{L}(\theta_{t-2}) \cdot m_{t-1} + \beta(\nabla\mathcal{L}(\theta_{t-1}) - \nabla\mathcal{L}(\theta_{t-2})) \cdot m_{t-1}] \\
&\geq \mathbb{E}[(1-\beta)\|\nabla\mathcal{L}(\theta_{t-1})\|_1 + \beta\nabla\mathcal{L}(\theta_{t-2}) \cdot m_{t-1} - 3\|L\|_1\beta],
\end{aligned}$$

where the last inequality follows by Hölder's inequality and Lemma 1. Applying the inequality until $t = 1$ and using the initialization $m_0 = 0$,

$$\begin{aligned}
\mathbb{E}[\nabla\mathcal{L}(\theta_{t-1}) \cdot m_t] &\geq \mathbb{E}[\beta\nabla\mathcal{L}(\theta_{t-2}) \cdot m_{t-1} + (1-\beta)\|\nabla\mathcal{L}(\theta_{t-1})\|_1 - 3\|L\|_1\beta] \\
&\geq \mathbb{E}[(1-\beta)\sum_{k=1}^{t}(\beta)^{t-k}\|\nabla\mathcal{L}(\theta_{k-1})\|_1 - 3\|L\|_1\sum_{k=1}^{t}\beta^{t-k+1}] \\
&\geq \mathbb{E}[(1-\beta)\sum_{k=1}^{t}(\beta)^{t-k}\|\nabla\mathcal{L}(\theta_{k-1})\|_1 - \frac{3\beta\|L\|_1}{1-\beta}] \\
&\geq \mathbb{E}[(1-\beta)\|\nabla\mathcal{L}(\theta_{t-1})\|_1 - \frac{3\beta\|L\|_1}{1-\beta}].
\end{aligned}$$

Thus,

$$\frac{1}{\sum_{t=1}^{T}\eta_t}\sum_{t=1}^{T}\eta_t\mathbb{E}[\|\nabla\mathcal{L}(\theta_{t-1})\|_1] \leq \frac{\mathcal{L}(\theta_0) - \mathcal{L}^\star}{(1-\beta)\sum_{t=1}^{T}\eta_t} + \frac{(1+5\beta)\|L\|_1}{2(1-\beta)^2}.$$

$\square$

### C.3 Proof of Lemma 1

*Proof.* Let $s = \text{sign}(\nabla\mathcal{L}(x + \Delta) - \nabla\mathcal{L}(x))$. Hence, from the smoothness assumption,

$$\mathcal{L}(x + s) - \mathcal{L}(x) - \nabla\mathcal{L}(x) \cdot s \leq \frac{1}{2}\|\boldsymbol{L}\|_1,$$

$$-\mathcal{L}(x + s) + \mathcal{L}(x + \Delta) + \nabla\mathcal{L}(x + \Delta) \cdot (s - \Delta) \leq \frac{1}{2}\sum_{i=1}^{d}L_i(s_i - \Delta_i)^2 \leq 2\|\boldsymbol{L}\|_1,$$

and

$$\mathcal{L}(x) - \mathcal{L}(x + \Delta) - \nabla\mathcal{L}(x + \Delta) \cdot (-\Delta) \leq \frac{1}{2}\|\boldsymbol{L}\|_1.$$

Summing the inequalities,

$$(\nabla\mathcal{L}(x + \Delta) - \nabla\mathcal{L}(x)) \cdot s \leq 3\|\boldsymbol{L}\|_1.$$

$\square$

## D Synthetic experiments

In this section, we study synthetic linear regression tasks with both separable and non-separable objectives. These experiments are designed to illustrate the behavior of our method under mini-batch optimization, where the gradient sign is affected by sampling noise.

Our setup uses a dataset $X \in \mathbb{R}^{n\times d}$ containing $n = 1024$ samples of dimension $d = 128$. Ground-truth predictions $y \in \mathbb{R}^n$ are generated by first sampling a ground-truth solution $\theta^\star \in \mathbb{R}^{d\times 1}$ and then setting

$y = X\theta^\star$. The loss of a parameter vector $\theta \in \mathbb{R}^d$ associated with a row $x_i$ of $X$ is given by $\mathcal{L}_i(\theta) = \frac{1}{2}(\theta \cdot x_i - y_i)^2$.

We consider the following two tasks:

1. **Separable objectives.** To construct a multi-sample separable objective, we proceed as follows. The matrix $X$ is constructed to have orthonormal columns by taking the $Q$ matrix from a QR decomposition of a random matrix with i.i.d. normal entries. To account for different coordinate scalings, we multiply $Q$ by $\sqrt{n/d}\,\mathrm{diag}(\sqrt{s}) \in \mathbb{R}^{d \times d}$, where $s$ is sampled from an exponential distribution with unit scale and the square root is applied element-wise. In this case, the objective satisfies

$$\mathcal{L}(\theta) \triangleq \frac{1}{n}\sum_{i=1}^n \mathcal{L}_i(\theta) = \frac{1}{2n}\|X\theta - X\theta^\star\|^2 = \frac{1}{2n}(\theta - \theta^\star)^\top X^\top X(\theta - \theta^\star)$$

$$= \frac{1}{2d}(\theta - \theta^\star)^\top \mathrm{diag}(s)(\theta - \theta^\star) = \frac{1}{2d}\sum_{i=1}^d s_i(\theta_i - \theta_i^\star)^2.$$

   To match Theorem 1, we use $r_{\min} = -1$ and $r_{\max} = 1$, and apply clipping. The initial weights are sampled uniformly from $\{-1, 0, 1\}^d$, while the ground-truth vector is sampled uniformly from $[-1, 1]^d$.

2. **General linear regression.** In this setting, the entries of $X$ are sampled independently from a normal distribution and scaled by $1/\sqrt{d}$. To match Theorem 2, we use $r_{\min} = -\infty$ and $r_{\max} = \infty$, and perform no clipping. The initial weights are sampled uniformly from $\{-3, -2, -1, 0, 1, 2, 3\}^d$, while the ground-truth vector is sampled uniformly from $[-3, 3]^d$.

Optimization is performed using parameters $\beta = 0.9$ and $\eta_t = 0.75/t$, matching the setup in Section 6. We run the algorithm for $T = 2048$ optimization steps with varying mini-batch sizes. All experiments are conducted using ternary gradients (obtained by applying the sign operator to the mini-batch gradient).

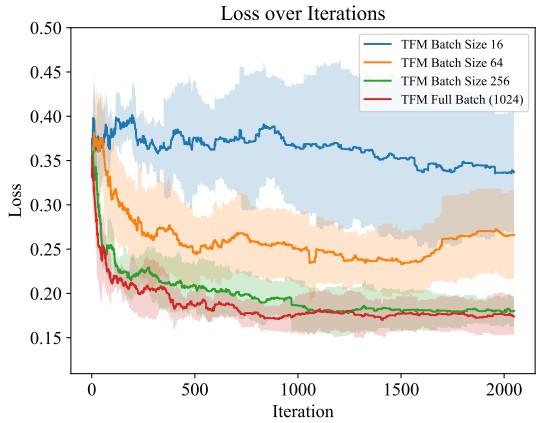
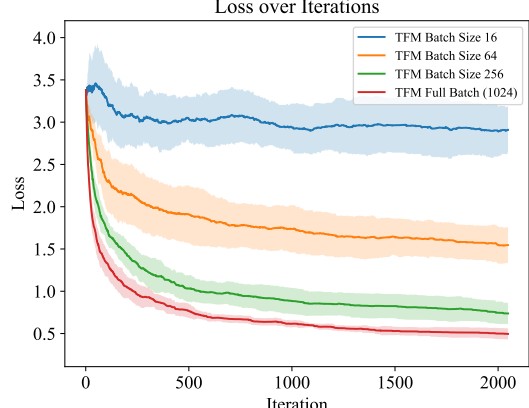

(a) Separable objective with ternary weights.

(b) General linear regression without clipping.

Figure 8: Loss of Ternary First Momentum (Algorithm 1) on synthetic linear regression with different batch sizes.

Fig. 8 illustrates the convergence behavior of Algorithm 1 under different batch sizes, which induce varying levels of noise in the ternary gradients. Full-batch optimization yields the best performance, with consistent improvement over the iterations considered. Nevertheless, we observe a decrease in loss across iterations even when using mini-batches. In the separable case, a batch size of 256 achieves performance comparable

to full batch, whereas it degrades more noticeably in the general linear regression setting. Smaller batch sizes of 64 and especially 16 lead to substantial degradation in performance.

These observations indicate that gradient noise plays a crucial role in the optimization of Algorithm 1, highlighting the importance of stable ternary optimization.

