# OpenReview forum: "Ternary Momentum For Quantized Training"
_TMLR — Accepted by TMLR_

### Review · Reviewer_adFL · 2025-12-08

**Summary Of Contributions:**

The paper presents an optimization approach, based on sampling Bernoulli masks, where the values of both the parameters and a momentum state are kept ternary. The motivation is training on resource-constrained devices.
The paper presents experiments that show some gain compared to omitting the momentum ingredient from the optimizer, and some theoretical results under simplifying assumptions.

**Audience:**

Yes

**Audience Explanation:**

The work may be relevant to audience interested in training networks on resource-limited devices.

**Claims And Evidence:**

No

**Claims Explanation:**

Following are my comments regarding the proposed method.

1. What is the precision of the model's activations?
The memory-efficient motivation of the paper suggests that it should be taken into account, but this aspect seems to be overlooked.

2. It is not clear to me how to draw Bernoulli masks of arbitrary real p \in (0,1) (namely, beta and eta_t in the context of the paper) without high precision computation or high computational complexity (e.g., loop) per entry of the Bernoulli mask. Please elaborate on this.
As the masks are computed per optimization update, what is precisely the complexity gain compared to quantizing the parameters after each update?


3. The assumption in setup 1 of the theoretical analysis, namely that the loss function is separable with respect to the parameters, is not practical for nonlinear models. This should be emphasized.
The assumption in setup 2 of the theoretical analysis, namely that no clamping is done, leads to the fact that each entry of theta_t can have any positive/negative integer value. This is in contrast with the motivation for ternary parameters, and should be emphasized as well.

4. Please report training time and present training loss curves for the proposed approach with and without momentum and for plain full precision training.

5. I would also like to see comparison with non-sampling based optimizers where the parameters/momentum are quantized to ternary values after each step.

6. In the fine-tuning experiments you use pertained networks and tune only LoRA. I would like to see an examination of the approach after the pretrained models are quantized.
Otherwise, what is the storage gain during training/inference? You will still need to store many full precision parameters/activations. Present the parameter count for these setups (as done in Table 1 for other setups).


7. Given the memory-efficient motivation of the work, it would be interesting to see the performance of the models when the activations are being quantized as well. How do they perform compared to methods that take the quantization of both the weights and the activations into account during training?


8. In some setups (not that few, in both Tables 1 and 3) there is only minor/negligible gain from using the ternary momentum compared to not using it. Elaborate on configurations of the setups under which the gain is higher/lower.

**Requested Changes:**

Please address the above comments.

---

> ### Author Response · Authors · 2025-12-31
> **Reply to Reviewer adFL**
>
> We thank the reviewer for the detailed feedback and valuable suggestions.
>
> ---
>
> ### (1, 6, 7) Memory efficiency and quantization of weights and activations
>
> We have added experiments with pretrained weight quantization (to 8 bits) and 8-bit activation quantization (see Tables 4 and 5 in the revised paper). Results show that ternary momentum can be effectively combined with pretrained weight quantization and activation quantization, achieving memory savings while preserving accuracy. This demonstrates the method’s applicability in memory-constrained settings, confirming that the proposed approach is compatible with broader memory-reduction strategies.
>
> ---
>
> ### (2) Bernoulli mask sampling and computational complexity
>
> We agree that sampling Bernoulli masks with arbitrary probabilities $\( p \in (0,1) \)$ requires high-precision operations. In practice, the memory and computation overhead is minor, as these operations are temporary for each update and need not be stored. Masks can also be sampled layer-wise, further reducing overhead. Compared to full-precision optimizers with momentum, which require storing full-precision optimizer states and repeated parameter quantization, our method applies updates directly in the ternary domain, avoiding extra operations between full-precision and ternary states.
>
> ---
>
> ### (3) Theoretical assumptions
>
> We have emphasized in Section 5 that Assumptions 1 and 2 are restrictive. Such simplifications are necessary to obtain rigorous convergence guarantees in the ternary and sign-based setting. Despite the gap with practical nonlinear models, the analysis provides principled insight into algorithm behavior under tractable conditions.
>
> ---
>
> ### (4) Training time and loss curves
>
> We now include training loss curves for ternary optimization with and without momentum. The results confirm that incorporating ternary momentum accelerates convergence across tasks. These plots are included in the appendix (see Figure 4).
>
> ---
>
> ### (5) Comparison with non-sampling ternary updates
>
> We added Table 8 comparing non-stochastic updates on the IMDB dataset, where momentum and parameters are quantized to ternary values after each step. The results show that while accuracy gains are observed in some configurations, benefits are inconsistent across gradient approximations, and training instability arises in certain setups, indicating that additional algorithmic modifications may be required for reliable convergence.
>
> ---
>
> ### (8) Variation of gains across setups
>
> The magnitude of improvements with ternary momentum varies across tasks and gradient approximations. In some setups, gains are minor, likely due to high stochasticity in the gradient approximation or suboptimal hyperparameter choices. For details and discussion of specific configurations, we refer the reviewer to the Results section of the revised manuscript.

---

### Review · Reviewer_5N4S · 2025-12-11

**Summary Of Contributions:**

Summary of Contributions:
This paper introduces a stochastic ternary momentum mechanism for training neural networks under extreme quantization, where parameters, gradients, and optimizer states are restricted to the ternary set {-1,0,+1}. The method replaces continuous momentum with a coordinate-wise Bernoulli masking rule that, in expectation, recovers the behavior of heavy-ball momentum while using only integer-valued operations. The authors provide convergence results for both convex and smooth non-convex objectives, and present experiments on IMDB, ImageNette, and ternary LoRA finetuning of large language models. Across these settings, ternary momentum consistently outperforms no-momentum baselines and approaches the performance of full-precision momentum with significantly reduced optimizer memory.

Key strengths:
1. The paper proposes a clean and intuitive discrete momentum mechanism that maintains the expected behavior of classical momentum while operating entirely in the ternary domain.
2. The update rule requires only integer arithmetic and Bernoulli masks, making it conceptually appealing for low-precision or hardware-constrained training environments.
3. Theoretical results cover both convex and smooth non-convex scenarios, providing justification that the proposed stochastic ternary updates remain stable and converge to approximate minima or stationary points.
4. The empirical evaluation spans multiple architectures and tasks, including MLPs, ViTs with ternary components, and ternary LoRA finetuning of LLMs, demonstrating consistent improvements over no-momentum baselines and competitive performance relative to full-precision momentum.

Key weaknesses:
1. Although optimizer memory is a primary motivation, the paper does not provide detailed system-level measurements (e.g., end-to-end memory usage or runtime behavior), so the practical impact remains somewhat unclear.
2. Experimental settings are relatively small in scale; results on industrial-scale or pretraining-scale workloads would strengthen the claims about practicality.
3. The theoretical assumptions (e.g., separability or smoothness) do not fully capture modern deep neural network training dynamics, leaving a gap between the theoretical guarantees and real-world practice.
4. The method is not compared against stronger low-precision optimizer baselines (such as 8-bit or 4-bit Adam), which makes it harder to position ternary momentum within the broader landscape of quantized optimization methods.

**Audience:**

Yes

**Audience Explanation:**

The paper addresses a focused but relevant problem in low-precision and memory-efficient training, an area that has an active community within TMLR’s readership. Researchers working on quantized optimization, hardware-aware training algorithms, or resource-constrained deep learning would likely be interested in the proposed ternary momentum mechanism and its theoretical and empirical findings.

**Broader Impact Concerns:**

This work focuses on an optimization algorithm for low-precision training and does not raise clear ethical or societal risks. The method is technical in nature and does not involve sensitive data, deployment decisions, or applications with direct human impact. A short statement clarifying this, along with noting that more efficient training may indirectly reduce energy usage, would be sufficient. No major broader impact concerns arise from the current submission.

**Claims And Evidence:**

Yes

**Claims Explanation:**

The main claims are reasonably well supported. The ternary momentum update is clearly specified, and its connection to heavy-ball momentum in expectation is mathematically justified. The convergence results, under standard smoothness assumptions, are consistent with the proposed algorithm. Experiments on IMDB, ImageNette, and ternary LoRA finetuning show consistent gains over no-momentum baselines and competitive performance relative to full-precision momentum, supporting the core claims. Some aspects (e.g., large-scale system measurements and comparisons to stronger low-precision optimizers) are less thorough, but this does not undermine the primary conclusions.

**Requested Changes:**

1. Provide more detailed reporting of the optimizer memory savings in practice. Since memory efficiency is a primary motivation, adding clearer measurements (e.g., total training memory usage with and without ternary momentum) would strengthen the empirical section.

2. Include a brief comparison or discussion relating the proposed method to existing low-precision optimizers such as 8-bit or 4-bit Adam. This would help situate ternary momentum more clearly within the broader landscape of memory-efficient optimization.

3. Clarify the experimental setup for the ternary gradient approximations (e.g., TernGrad, sparse ZO, No-Backward). A short explanation of how each surrogate gradient is computed would make the experiments easier to interpret.

4. Add a brief discussion about limitations, such as the scalability of the method to very large-scale pretraining settings or potential issues with highly noisy ternary gradients. This would improve clarity without requiring additional experiments.

---

> ### Author Response · Authors · 2025-12-31
> **Reply to Reviewer 5N4S**
>
> We thank the reviewer for their careful reading and constructive feedback.
>
> ---
>
> ### (1) Optimizer memory and practical impact
>
> We have added Table 7, which reports the peak memory usage during training. While end-to-end memory is sometimes dominated by activations and model parameters, and therefore may not change dramatically in all settings, we observe substantial and consistent savings in optimizer-state memory (see also Table 2), which is the primary target of our method.
> Our experiments are mainly reported to support the optimization performance of our method in constrained environments rather than to emphasize memory savings. We additionally include results for ternary LoRA fine-tuning, where the original weights, which constitute the dominant memory cost, are themselves quantized. In this regime, our optimizer yields accuracy benefits (Table 5). These results support the view that the proposed method is complementary to existing memory-reduction techniques.
>
> ---
>
> ### (2) Limitations and scalability
>
> We have added a limitations paragraph in Section 7, discussing scalability to very large-scale pretraining and clarifying that the method is not intended to replace full-precision industrial-scale pretraining.
>
> ---
>
> ### (3) Theory vs. practice
>
> We agree that a gap remains between the theoretical assumptions and modern deep network training dynamics. As is common in optimization work motivated by deep learning (e.g., analyses of SGD variants or Adam), simplified settings are necessary to obtain rigorous guarantees. Our analysis aims to provide principled insight under tractable assumptions for the highly restrictive and largely unexplored setting of bounded-integer learning, which inherently requires stronger conditions.
>
> ---
>
> ### (4) Comparison with low-precision Adam variants
>
> We intentionally refrain from including quantized Adam baselines because they are not directly compatible with our setting. Our method assumes access only to ternary gradients, under which second-moment statistics become degenerate (since \((-1)^2 = 1^2 = 1\)), and meaningful Adam-style updates would require additional algorithmic modifications. Moreover, existing 8-bit and 4-bit Adam methods typically optimize full-precision parameters, which deviates from the fully ternary regime considered in this work. We now clarify this distinction in the paper and leave such extensions to future work.
>
> ---
>
> ### (5) Clarification of experimental setup
>
> We have added a concise description of the surrogate gradient constructions (TernGrad, sparse ZO, No-Backward) in the experimental section to improve interpretability.

---

### Review · Reviewer_nAXh · 2025-12-19

**Summary Of Contributions:**

The paper proposes a simple ternary momentum mechanism that maintains a ternary first-moment state and applies stochastic updates whose expectation mimics classical momentum whle avoiding FP optimizer state. It argues that this yields substantial theoretical reductions in optimizer-state memory and provides convergence to a neighborhood guarantees under idealized assumptions with exact sign gradients. Empirical results suggest that ternary momentum can improve stability and performance over no-momentum baselines in quantized training settings albeit the experiments primarily illustrate feasibility rather than tightly validating the theory.

Strengths
- Good, simple optimizer idea. Intuative and lightweight way to reintroduce momentum behavior while keeping the state ternary.
- Theoretical analysis with nontrivial guarantees
- Empirical breadth coudl be improved but evaluates across multiple tasks/settings (from scratch and finetuning), showing the method can be competitive relative to simple baselines.
- A relatively clear systems motivation focusing on optimizer related memory (a bottleneck in large-scale training).

Weaknesses
- Theory vs. practice gap. Theory assumes access to exact sign/ternary gradient information and idealized storage, while several experimental gradient approximations may rely on FP components or overhead (this seems standard in the field but remains a key issue for full impact of the papers ideas).
- Baseline/attribution clarity: comparisons and “following prior work” framing would benefit from more explicit mapping of baselines to specific prior algorithms and fuller metric alignment.
- Limited ablations/robustness reporting: sensitivity to factors like batch size/gradient noise and variability across runs is not deeply explored in the current presentation.

**Audience:**

Yes

**Audience Explanation:**

The paper addresses memory efficient optimization via ternary momentum, a topic relevant to readers interested in quantized training, optimization methods, and efficient learning. Its combination of algorithmic ideas, theoretical analysis, and empirical evaluation would - subject to the suggested corrections - be of interest to a subset of TMLR’s audience, particularly those working on low-precision and resource-constrained training.

**Claims And Evidence:**

No

**Claims Explanation:**

Overall, while I think the paper proposes an interesting idea and provides relevant theory, I find the current paper suffers from an underdeveloped experimental evaluation, lack of deep insights into the (practical) properties of the proposed algorithm, a gap between theoretical assumptions and practical implementation and thus practical relevance. Addressing the major concerns below would be necessary to provide enough insights to backup the general claims of relevance and correctness.


1. Redundancy and structure (Sections 1-4) - Moderate concern:
While the introduction is generally clear, I feel it contains substantial redundancy across Sections 1-4. In particular, the three long summary bullet points largely restate issues already discussed earlier in the text, resulting in an unnecessarily verbose presentation. The introduction would benefit from restructuring and consolidation; it feels overly long relative to its informational content.

2. Lack of a synthetic experiment validating theory and demonstrating properties - Major concern:
The paper presents non-trivial theoretical results, yet misses an opputunity to provide a (potentially synthetic) optimization experiment designed to validate these results under the stated assumptions. A controlled synthetic experiment would be helpful to demonstrate that the algorithm behaves as predicted by the theory (see also 2b) and to clarify the relevance of the theoretical analysis to the proposed method. Without this, the theory remains largely disconnected from the empirical evaluation (e.g. the theory assumes access to determistic gradients, but the experiment do not).

2. Batch size and gradient noise are insufficiently explored - Major concern:
Ternary and sign-based optimization methods are well known to be sensitive to gradient noise, making batch size a critical factor. Although briefly mentioned, this issue is neither analyzed nor experimentally evaluated. The absence of a systematic discussion or ablation significantly weakens the empirical section. Targeted experiments (e.g., low-dimensional or synthetic settings) woudl be very helpful to assess stability and behavior across batch sizes.

3. Weak justification of experimental use cases - Major concern
I did not find the experimental choices convincingly motivated specifically for studying ternary momentum/ternary optimization. The paper often relies on “following prior work” (e.g., Di Castro et al.) rather than clearly stating what each setting is intended to test or why it is particularly diagnostic for the proposed method (e.g., memory constraints, sign noise, stability). As a result, the experimental section feels under-contextualized and would benefit from clearer motivation and framing.

4. Baseline positioning and comparison to prior work - Major concern:
As far as I can tell, the baselines are not clearly and explicitly mapped to specific prior methods/papers, making it difficult to assess novelty and progress relative to existing literature. Explicitly identifying which algorithms or papers each baseline corresponds to is necessary for easy comparison.

5. Theory–practice mismatch and memory claims - Major concern:
The narrative surrounding the memory efficiency claims is a bit confusing to me. The theoretical analysis assumes idealized storage (e.g., log2(3) bits per parameter with no overhead), whereas the experimental setups necessarily has memory overhead and in some cases even rely on full-precision gradients. Even highly optimized implementations would require at least two bits as far as I know. While the paper acknowledges some of these limitations, the implications are not sufficiently reflected in the framing of the results, which appear closer to simulations of an idealized regime than realizable systems - perhaps the authors can help me to better understand this slightly unusual approach (as an outsider to the ternary research area).

6. Missing variability and statistical reporting - Moderate concern:
Tables 1 and 3 report single accuracy values without any measure of variability (e.g., standard deviation, confidence intervals, or multiple runs). Given the modest effect sizes and the stochastic nature of the methods, this omission makes it difficult to assess robustness or statistical significance.

7. Omission of SparseZO in Table 1 - Moderate concern:
SparseZO is discussed in the paper but is not included in Table 1 (only Table 3), without explanation. This omission is confusing and I recommend clarifying and ideally including it.

8. Incomplete alignment with Di Castro et al.'s metrics - Moderate/Major concern :
The paper claims to follow the experimental design of Di Castro et al., yet omits several metrics used in that work (e.g., number of updates and other efficiency-related measures). To me, this weakens the comparability of results and makes the “following” claim questionable. Either the framing should be adjusted, or idealy the missing metrics should be included.



9. List of minor comments:
 - p2, mid $r_{max=1}$-> $r_{max}=1$
 - log2(3) perhaps spell out where this comes from
 - p4. I’d strongly suggest providing references inline for the claims made in “Optimization without momentum” paragraph.
 - p5, last paragraph. “motivate” -> “motivated”
 - p9: “that is possible acquired in an efficient way.” -> wording
 - Table 3: Please include the actual metric and preferably also if higher is better or lower is better.
 - Table 3: For consistency, I’d suggest having the same number of decimals for the three fine-tuning tasks.

**Requested Changes:**

See the list of concers above (items listed as Major are items I feel the authors need to address)

---

> ### Author Response · Authors · 2025-12-31
> **Reply to Reviewer nAXh**
>
> We thank the reviewer for the careful reading and constructive feedback. We have substantially revised the manuscript to address the concerns raised.
>
> ---
>
> ### (1) Redundancy and structure (Sections 1–4)
>
> We removed redundant material from the Related Work (Section 2) and the method section (Section 4) by replacing long bullet lists with concise summaries of the method’s key benefits. The Related Work section has been streamlined to focus more clearly on quantization and memory-efficient optimization.
> The Introduction is largely kept unchanged, as we believe it already clearly states the problem setting, positions our contribution with respect to prior work, and motivates the paper’s focus effectively.
>
> ---
>
> ### (2 & 3) Lack of synthetic validation and batch-size / noise sensitivity
>
> We added a new synthetic experiment in Appendix D, demonstrating convergence behavior under the theoretical assumptions for separable and logistic regression problems. We further evaluate the impact of mini-batch size on convergence. As expected, larger batch sizes reduce noise and yield lower final loss, while smaller batches exhibit higher variance.
>
> ---
>
> ### (4 & 5) Experimental motivation and baseline positioning
>
> We reorganized and clarified the experimental setup to better motivate each evaluation. The various gradient approximation methods are not baselines themselves, but are used to test the robustness of ternary momentum under different ternary gradient approximations. We now explicitly explain the role of each approximation scheme and why the Di Castro et al. framework is particularly suitable for our optimizer.
>
> Regarding baselines, we clarify that there are no off-the-shelf optimizers designed for the fully ternary setting considered here. Consequently, we use full-precision momentum with stochastic updates and no-momentum variants as the most appropriate reference points.
>
> ---
>
> ### (6) Theory–practice mismatch and memory framing
>
> We clarify the origin of the $\(\log_2(3)\)$ term and explicitly discuss the idealized nature of the theoretical memory analysis. We further note that ternary memory requires additional practical overhead, and we include this limitation explicitly in Section 7.
>
> ---
>
> ### (7) Variability and statistical reporting
>
> We now report standard deviations for the results in Tables 1 and 3, enabling assessment of robustness and statistical reliability.
>
> ---
>
> ### (8) Omission of Sparse ZO from Table 1
>
> Sparse ZO is evaluated only in the fine-tuning setting (Table 3), following the experimental protocol and codebase of the Zero-Order Benchmarking framework, which is specifically designed for large language model fine-tuning. Table 1 corresponds to training-from-scratch experiments conducted in a different pipeline. Including Sparse ZO there would require substantial re-engineering, which we believe is outside the scope of this paper.
>
> ---
>
> ### (9) Alignment with Di Castro et al.’s metrics
>
> We now report the expected number of parameter and optimizer-state changes, together with the corresponding expected energy consumption. Since Di Castro et al. do not use optimizer states, their methods naturally incur fewer changes. Moreover, in our method the reported parameter changes constitute an upper bound, as updates are mediated by the momentum mechanism and learning rate rather than applied directly at each step.
>
> ---
>
> ### (10) Minor corrections
>
> We addressed all listed minor comments, including clarifying the origin of $log_2(3)$, adding inline references in the “Optimization without momentum” paragraph, correcting typos and wording, standardizing decimal precision in Table 3, and explicitly stating the evaluation metric and its direction.

---

> > ### Comment · Reviewer_nAXh · 2026-01-30
> >
> > Apologies for the slightly delayed response.
> >
> > I thank the authors for the extensive revisions and the additional experiments, which have helped me better understand the proposed method.
> >
> > I have a few remaining comments and questions that I would appreciate the authors addressing before I make a final recommendation.
> >
> > - Tables 1 and 3 (statistical reporting): It is unclear what the reported “±” values represent. Please clarify this explicitly in the table captions. In addition, please clarify the criterion used for boldfacing results (e.g., simply the highest value or only statistically significant improvements).
> >
> > - Figure 4 and optimisation behaviour: In my view, the benefits of ternary momentum-  and optimisation methods more generally - are most clearly illustrated by training traces such as those shown in Figure 4. I believe such figures could (and perhaps should) be included in the main text rather than relegated to the appendix.
> > Relatedly, Figure 4 does not include a full-precision baseline without momentum, which would help illustrate how momentum influences the ternary setup vs the FP setup. Moreover, the loss function corresponding to the y-axis is not specified. I am also slightly concerned about the observed behaviour of the loss traces: on IMDB, the loss gap between FP and ternary variants appears quite large while the final accuracies are very similar. Is this behaviour expected on IMDB (it could be the case, I am just surprised)? I would also find it useful to see analogous curves on the validation set (and possibly the test set), and potentially plots in terms of accuracy rather than only training loss, to be fully convinced.
> >
> > - The new results in Appendix D help connect the empirical behaviour to the theoretical analysis and, as far as I can tell, do support the theory under its intended assumptions (including the role of batch size and reduced gradient noise). However, this connection is currently undercommunicated in the main text. I would suggest explicitly briefly discussing these results in the main paper to make the theory–experiment link more visible to readers who may not study the appendix in detail.
> >
> > Minor / nitpicking:
> >
> > - Table 1: The placement of the value “90.8” is visually confusing and makes it unclear which column it corresponds to.
> >
> > - Tables 1 and 3 (layout): I recommend reconsidering the table layout, in particular by adding a clear heading for the first column (e.g., “Optimizer” or “Method”). This would make it easier to map rows directly to the methods and variants outlined in the text.
> >
> > - Definition of “FP” in Tables 1 and 3: After rereading the paper, I became unsure what exactly “FP” refers to in these tables, particularly whether momentum is included or not. Since momentum is explicitly distinguished for other methods, I suggest being more explicit - both in the text and in the table captions.
> >
> > - Placement of tables: Please ensure that tables are placed close to the corresponding experimental setup descriptions to improve readability.
> >
> > - Tables 4 and 5: Strictly speaking, the evaluation metric is not clearly specified in the captions. Please add this information for completeness.

---

> > > ### Author Response · Authors · 2026-02-08
> > > **Response to Reviewer Comments**
> > >
> > > We thank the reviewer for their thoughtful comments and engagement. We apologize for the delayed response and appreciate the opportunity to clarify these points.
> > >
> > > ### 1. Statistical Reporting (Tables 1 and 3)
> > >
> > > **STD Clarification:** We have updated the captions for Tables 1 and 3 to explicitly state that the reported “$\pm$” values represent the standard deviation.
> > >
> > > **Boldfacing:** We have clarified in the text that boldface indicates the highest average accuracy within each baseline group.
> > >
> > > ### 2. Optimization Behavior and Loss Traces (Figure 4)
> > >
> > > **Placement:** Per your suggestion, we have moved the loss curve figure to the main body of the paper to better highlight the optimization benefits.
> > >
> > > **Evaluation Loss:** We mention that the loss we report and train with is cross-entropy loss.
> > >
> > > **FP with Momentum:** In the main body of the paper we include FP training curve with AdamW since this is the standard practice for FP training. In Figure 5 in the appendix we isolate the influence of momentum on ternary vs. FP setups. We find that while momentum accelerates ternary convergence, the FP case reaches a lower final loss, though at a similar convergence rate.
> > >
> > > **The Loss Gap (IMDB):** We have included Figure 6, which details test loss curves. The gap between ternary and FP variants is significantly smaller with the test set than with the training set.
> > >
> > > **Loss Gap Explanation:** The large training loss gap is expected in quantized training. In FP, the model can minimize Cross-Entropy ($L = -\sum y_i \log p_i$) toward zero by indefinitely increasing logit magnitudes to maximize confidence. In ternary training, the weights are restricted to $\{-1, 0, 1\}$, creating discretization noise. The model successfully learns the correct decision boundaries (preserving accuracy), but the weight constraint prevents it from reaching the same low loss value as in FP.
> > >
> > > ### 3. Theory–Experiment Link
> > >
> > > **Main Text Integration:** We have included a brief summary of the simulation results from Appendix D in the main body. This explicitly connects the empirical behavior (role of batch size and reduced gradient noise) to our theoretical analysis, making the theory-experiment link more accessible to the reader.
> > >
> > > ### 4. Minor Corrections and Layout
> > >
> > > **Table Readability:** We improved tables’ readability with added headers, improved captions, and structural adjustments.
> > >
> > > **Tables 4 & 5:** Captions now explicitly specify the evaluation metric used.
> > >
> > > **Proximity:** Tables have been moved closer to their respective descriptions to improve the flow of the manuscript.

---

### Author Response · Authors · 2025-12-31
**Author Response & Changes**

We thank the reviewers for their thoughtful and constructive feedback.  In response, we have substantially revised the manuscript, including:

- Expanded experimental evaluation with additional quantization of activations and pre-trained weights, and a new non-stochastic ternary momentum baseline (Tables 4, 5, 8).
- Improved presentation and organization, especially in Sections 2, 4, and 6.
- Clarified assumptions limitations - theoretically (Section 3) and we explicitly discussed limitations (Section 7).
- Added detailed analysis of memory consumption and expected energy usage (Tables 6, 7).
- Added synthetic experiments (Appendix D) and new batch-size ablations.

We believe these changes address the concerns raised by all reviewers and substantially strengthen the paper. We are happy to engage in further discussion and provide any clarification needed.

---

### Decision · Action_Editor_26mD · 2026-02-23

**Recommendation:** Accept with minor revision

**Additional Comments:**

Thanks to the authors for diligently engaging in the discussion and revising the paper.

I have two additional requests; they are optional, but I would strongly appreciate it if the authors could go some extra miles to discuss these points and provide some experimental results:

1. As pointed out by all reviewers, there is currently a large theory vs. practice gap, i.e., a big gap compared to full-precision training (as illustrated in Figure 1 and Figure 5 in the paper). I think the problem is partially because the kind of worst-case analysis of the convergence bound does not really capture the theoretical nature of gradient noise -- Do we know why there is a big gap compared to full-precision training (we know intuitively it's probably because of the gradient noise, but can the current theory capture this intuition in the convergence bound)? Can we expect advancing algorithms to eliminate the gap one day, or are there fundamental gaps that are theoretically irreducible?

It is noteworthy that we already have a good theory on the average-case behavior of SGD (https://arxiv.org/abs/2102.04396) and SGD with momentum (https://arxiv.org/abs/2106.03696), that can accurately calculate gradient noise. At least for linear regression models, the loss-over-iterations curve of SGD and momentum (with full precision) can be completely predicted. It seems rather straightforward to me to consider low-precision updates under this perspective, and my hunch is that the kind of average-case analysis might be more useful theory.

Of course, I don't have answers to the above questions myself, and average-case analysis might be beyond the scope of this work. I would like to provide some empirical ideas as well:

 1a. The learning-rate schedule is set to `constant / t` throughout this work. Have you tried other learning-rate schedules? Because we know that a decaying learning-rate schedule can reduce gradient noise; can we expect a good learning-rate schedule, with possibly a larger number of iterations, to somehow improve the performance?

 1b. The momentum coefficient beta is set to constant in this work. But as pointed out by (https://arxiv.org/abs/2106.03696), such a heavy-ball momentum is essentially the same as SGD (under full precision) -- one has to use scheduled beta in order to make a difference. It might partially explain why the performance gain is marginal by introducing momentum in this work; can we achieve anything better with this theoretical insight?

2. Currently, all experiments in this work are "simulations", in the sense that they run on fully capable chips but just pretend that only ternary information is available. Do we have any concrete examples, of what a "resource-limited device" actually looks like? It could be only distantly related; but I would like to see at least some discussion towards this direction.

**Audience:**

Yes

**Audience Explanation:**

Yes. Improving training efficiency for large neural networks, possibly on resource-limited devices, would be a research topic of both theoretical and empirical interests.

**Claims And Evidence:**

Yes

**Claims Explanation:**

This is a solid work on quantization-preserving optimization, and the authors have addressed most of the reviewers' concerns.

---

> ### Author Response · Authors · 2026-03-27
> **Response to the Editor – Minor Revision**
>
> We thank the editor for the thoughtful suggestions and for the positive assessment of our work. We have addressed the points below and incorporated the corresponding clarifications and additions into the revised manuscript.
>
> ### 1. Theory–practice gap and the role of gradient noise
>
> In the revision, we expand the discussion in the experiments section to better contextualize the theory–practice gap. In practical settings with ternary quantization, there is an inherent approximation error, and the quantized optimization process introduces additional limitations. While improved algorithmic design may help reduce this gap, it is natural to expect a residual discrepancy compared to full-precision optimization.
>
> We further clarify in the theory section that our analysis focuses on a deterministic, worst-case setting, which does not capture the structure of stochastic gradient noise in deep models. We agree that average-case analyses could provide a more precise characterization of this gap, and we highlight this direction as future work.
>
> ### 2. Learning-rate schedules
>
> Following the editor’s suggestion, we extended our empirical study to include additional learning-rate schedules. The results are reported in Table 6 and Figure 5 (appendix). We observe that appropriate scheduling improves both convergence and final accuracy.
>
> ### 3. Momentum scheduling
>
> We agree that using a scheduled momentum parameter, rather than a constant one, may lead to more significant improvements. Incorporating such schemes into the quantized setting, as well as analyzing them theoretically, is an interesting direction that we leave for future work.
>
> ### 4. Resource-limited device considerations
>
> We added a discussion in the limitations section outlining practical considerations for deployment on resource-limited devices, including fully quantized forward and backward passes, hardware-aware architectural choices, and memory constraints.
>
> ---
>
> Overall, we believe these additions improve the clarity of the paper and better position it with respect to both theoretical and practical considerations. We thank the editor again for the helpful feedback.
>
> We apologize for the slight delay in submitting the revision due to the current situation in the Middle East, and we appreciate your understanding.

---

> > ### Comment · Action_Editor_26mD · 2026-03-30
> > **Thanks for the work.**
> >
> > Thanks to the authors for their diligent work, especially under the difficult circumstances.